EMBO
Molecular Medicine

# Antioxidant nanozyme counteracts HIV-1 by modulating intracellular redox potential

Shalini Singh[1,2], Sourav Ghosh[3], Virender Kumar Pal[1,2], MohamedHusen Munshi[2], Pooja Shekar[4], Diwakar Tumkur Narasimha Murthy[5], Govindasamy Mugesh[3,*] & Amit Singh[1,2,**]

## Abstract

**Reactive oxygen species (ROS) regulates the replication of human immunodeficiency virus (HIV-1) during infection. However, the application of this knowledge to develop therapeutic strategies remained unsuccessful due to the harmful consequences of manipulating cellular antioxidant systems. Here, we show that vanadium pentoxide ($V_2O_5$) nanosheets functionally mimic natural glutathione peroxidase activity to mitigate ROS associated with HIV-1 infection without adversely affecting cellular physiology. Using genetic reporters of glutathione redox potential and hydrogen peroxide, we showed that $V_2O_5$ nanosheets catalyze ROS neutralization in HIV-1-infected cells and uniformly block viral reactivation and replication. Mechanistically, $V_2O_5$ nanosheets suppressed HIV-1 by affecting the expression of pathways coordinating redox balance, virus transactivation (*e.g.*, NF-κB), inflammation, and apoptosis. Importantly, a combination of $V_2O_5$ nanosheets with a pharmacological inhibitor of NF-κB (BAY11-7082) abrogated reactivation of HIV-1. Lastly, $V_2O_5$ nanosheets inhibit viral reactivation upon prostratin stimulation of latently infected CD4[+] T cells from HIV-infected patients receiving suppressive antiretroviral therapy. Our data successfully revealed the usefulness of $V_2O_5$ nanosheets against HIV and suggested nanozymes as future platforms to develop interventions against infectious diseases.**

**Keywords** glutathione; glutathione peroxidase; HIV; latency; nanozymes
**Subject Categories** Microbiology, Virology & Host Pathogen Interaction; Pharmacology & Drug Discovery

## Introduction

Nanomaterials with intrinsic enzyme-mimetic properties (nanozymes) have been explored as low-cost alternatives to natural enzymes (Wei & Wang, 2013). The application of these nanomaterials was largely restricted to industries for chemical synthesis, detection of biomolecules, and bioremediation (de la Rica & Stevens, 2012; Wei & Wang, 2013). While largely ignored by the biomedical community, recent studies provide evidence for the clinical importance of artificial nanozymes *in vivo* (Salata, 2004; Das *et al*, 2007). For example, ceria-based nanoparticles (NPs) mimic superoxide dismutase (SOD) activity and exhibit neuroprotection and reduced inflammation (Korsvik *et al*, 2007). Similarly, iron oxide-based nanoparticles mimic peroxidase-like activity and protect from bacterial biofilms associated with oral infection (Gao *et al*, 2016). Moreover, ferumoxytol, an FDA approved iron oxide nanoparticle, has been shown to inhibit tumor growth in mice (Zanganeh *et al*, 2016). Recently, vanadium pentoxide ($V_2O_5$) nanomaterials were reported to mimic glutathione peroxidase (GPX)-like activity *in vitro* and protect mammalian cells from oxidative stress (Vernekar *et al*, 2014).

Our particular interest is to apply antioxidant nanozymes in the context of infection caused by human immunodeficiency virus (HIV-1; the causative agent of the acquired immunodeficiency syndrome [AIDS]; Perl & Banki, 2000). A major barrier to curing HIV-1 infection is latency, wherein the infected cells harbor the intact viral genome that is replication-competent but transcriptionally silent. Interestingly, oxidative stress is known to reactivate HIV-1 from latent reservoirs via NF-κB directed transcriptional activation of the viral long terminal repeat (LTR) (Staal *et al*, 1990; Pyo *et al*, 2008). Conversely, cellular antioxidant response along with the iron import pathway has been recently shown to promote the establishment of HIV latency (Shytaj *et al*, 2020). Further studies linking oxidative stress with HIV-1 infection demonstrate variations in glutathione (GSH) levels in infected cells and tissues (Buhl *et al*, 1989; Eck *et al*, 1989; Herzenberg *et al*, 1997). Using a non-invasive biosensor of GSH redox potential ($E_{GSH}$; Grx1-roGFP2), we discovered that reductive $E_{GSH}$ sustains viral latency, whereas a marginal oxidative shift in $E_{GSH}$ promotes HIV-1 reactivation (Bhaskar *et al*, 2015*).* The importance of reactive oxygen species (ROS) and GSH is evident from studies showing that molecules generating ROS or

---

1   Department of Microbiology and Cell Biology, Indian Institute of Science, Bangalore, India
2   Centre for Infectious Disease Research (CIDR), Indian Institute of Science, Bangalore, India
3   Department of Inorganic and Physical Chemistry, Indian Institute of Science, Bangalore, India
4   Bangalore Medical College and Research Institute, Bangalore, India
5   Department of Internal Medicine, Bangalore Medical College and Research Institute, Bangalore, India
    *Corresponding author. Tel: +91 8022933354; E-mail: mugesh@iisc.ac.in
    **Corresponding author. Tel: +91 8022933063; E-mail: asingh@iisc.ac.in

inhibiting GSH can be exploited to purge viral reservoirs (Savarino *et al*, 2009; Yang *et al*, 2009). Moreover, inhibition of another major antioxidant system—thioredoxin/thioredoxin reductase (Trx/TrxR) by auranofin selectively promotes differentiation and apoptosis of the memory CD4[+] T cells to eliminate HIV reservoirs (Chirullo *et al*, 2013). A combination of auranofin and antiretroviral therapy (ART) effectively reduced total viral DNA and integrated viral DNA in patients' peripheral blood mononuclear cells (Diaz *et al*, 2019).

Along with GSH and Trx/TrxR systems, altered levels of a major ROS detoxifying enzyme family—glutathione peroxidases (GPXs) are associated with HIV-1 reactivation and replication (Look *et al*, 1997a; Bhaskar *et al*, 2015). Interestingly, the HIV-1 genome also encodes a fully functional GPX (HIV-1 vGPX) module, which protects cells from ROS induced apoptosis and possibly helps HIV-1 to maintain latency (Zhao *et al*, 2000; Cohen *et al*, 2004). However, efforts to mitigate oxidative stress for subverting HIV-1 reactivation by either supplementation of GSH precursor (N-acetylcysteine [NAC]) or activation/over-expression of GPXs yielded inconsistent results (Sappey *et al*, 1994; Witschi *et al*, 1995; Sandstrom *et al*, 1998; De Rosa *et al*, 2000). While counterintuitive, these findings are in agreement with several studies showing the adverse influence of uncontrolled over-production of natural antioxidant systems (including GPXs) on redox metabolism and disease outcome (Sandstrom *et al*, 1998). Further, over-expression of GPXs might not be sufficient as GPX activity is enhanced by post-translational modifications (PTMs) such as phosphorylation, carbonylation, and O-GlcNAcylation (Cao *et al*, 2003; Yang *et al*, 2010; Wiedenmann *et al*, 2018). Besides this, the activity of GPXs is also dependent on selenium (Se), an essential micronutrient that has been reported to be low in HIV patients (Look *et al*, 1997a). In this context, artificial nanozymes mimicking GPX-like activity (*e.g.*, $V_2O_5$ NPs) with high sensitivity and specificity under physiological conditions found in the human body (*i.e.*, mild temperature, pH 4–8, and aqueous buffer) can provide a suitable alternate to natural GPXs. We envisage that antioxidant nanozymes can be exploited to generate new knowledge on redox signaling mechanisms underlying HIV-1 latency, which could aid the development of fresh therapeutic approaches for targeting HIV.

In this work, we explored the utility of $V_2O_5$ ultrathin nanosheet (Vs) in dissecting redox signaling underlying HIV-1 latency. By exploiting genetic reporters, cell lines, and primary cellular models of HIV-1 latency, we discovered that Vs-enabled remediation of ROS efficiently blocks reactivation and multiplication of HIV-1. Our study provides a first example of the therapeutic potential of antioxidant nanozymes against HIV.

## Results

### Synthesis and biophysical characterization of catalytically efficient $V_2O_5$ thin nanosheets

$V_2O_5$ nanomaterials are known to exhibit isoform-specific GPX activity, which is dependent on the surface exposed crystal facets (Ghosh *et al*, 2018). However, the biological usefulness of GPX activity associated with distinct morphologies of $V_2O_5$ nanomaterials against infectious disease remained untested. We morphed $V_2O_5$ nanomaterials into three discrete morphologies, i.e., nanowires (VNw), nanosheets (VSh), and ultrathin nanosheets (Vs), and confirmed by scanning electron microscopy (SEM) (Fig 1A–C). Next, we examined if these morphological changes exhibit new levels of functionality by measuring GPX activity. We measured GPX reaction kinetics by monitoring the decrease in NADPH absorbance at 340 nm using glutathione reductase (GR)-coupled assay (see schematic; Fig 1D). Examination of the rate of GPX activity revealed that Vs possesses a 1.6- to 2-fold higher capacity to reduce $H_2O_2$ in the GR-coupled assay as compared to VNw and VSh (Fig 1E). A comparison of activities with three different peroxides—$H_2O_2$, tertiary-butyl hydroperoxide (*t*-BuOOH), and cumene hydroperoxide (Cum-OOH), indicates that Vs is very selective toward $H_2O_2$ (Fig 1F). Importantly, while VNw and VSh were found to exert toxicity on a monocytic cell line (U1) latently infected with HIV-1 (Folks *et al*, 1987), Vs was well tolerated (Fig 1G). Similarly, Vs did not induce cytotoxicity in case of a lymphocytic cell line (J1.1) latently infected with HIV-1 (Appendix Fig S1A; Perez *et al*, 1991). Based on this, we selected catalytically efficient and non-toxic Vs nanozyme for extensive biophysical, biochemical, and biological characterization.

First, we recorded the crystalline nature of the lyophilized, thin nanosheets, Vs, by the powder X-ray diffraction pattern (PXRD) (Appendix Fig S2A). The PXRD pattern was indexed to the standard $V_2O_5$ orthorhombic phase (a = 11.5160 Å, b = 3.5656 Å, c = 4.3727 Å, JCPDS = 41–1,426, Space group Pmmn). Second, we examined the crystal facets exposed in the Vs material using high-resolution transmission electron microscopy (HRTEM) and selected area electron diffraction (SAED) pattern analysis. The observed lattice fringes for Vs are (200) and (110) with d-spacing of 0.58 nm and 0.34 nm, respectively, with the interfacial angle of 72.8°. The interfacial angle between two planes (110) and (−110) with equidistant d-spacing is 34.4° (Appendix Fig S2B and Fig 1H). These three planes in the HRTEM fall in the common zone axis [001]. In both the figures, SAED patterns (inset) were well indexed along [001] zone axis, which confirms that the surface exposed facets are indeed [001]. These results agree with the most intense peak due to (001) plane observed in the PXRD pattern of Vs.

Third, we performed FT-Raman spectroscopy to determine the nature of bonding formed between the metal and oxygen atoms in the orthorhombic $V_2O_5$ crystals. The FT-Raman spectra showed a peak around 995 cm$^{-1}$, which corresponds to the terminal (V = O) resulting from the unshared oxygen atom of the $V_2O_5$ crystal (Fig 1I). The peaks detected at lower vibration frequencies are consistent with the lattice vibrations of layered material (Sanchez *et al*, 1982; Zhou & He, 2008; Avansi *et al*, 2011). We examined the purity of Vs by confirming the detection signal for vanadium (V) and oxygen (O) using point energy-dispersive X-ray spectroscopy (EDS) (Fig 1J). A small signal detected at 2 KeV is due to the gold (Au) sputtering of the sample during spectrum recording. Fourth, we confirmed the elemental composition and purity of the Vs material by selected area bright field (SABF) images and X-ray mapping images (Fig 1K). Both these techniques confirmed that Vs material has a homogenous distribution of vanadium (V) and oxygen (O) (Fig 1, L and M). Finally, we determined the oxidation state of the Vs material using X-ray photoelectron spectroscopy (XPS). The analysis revealed binding energies (BE) and full width at half maxima (FWHM) for the V2p3/2 and V2p1/2 peaks as well as the difference in the BE between O1s and V2p3/2 orbitals (12.8 eV) (Fig EV1A–C). All of this confirms that vanadium exists in the + 5-oxidation state in Vs.

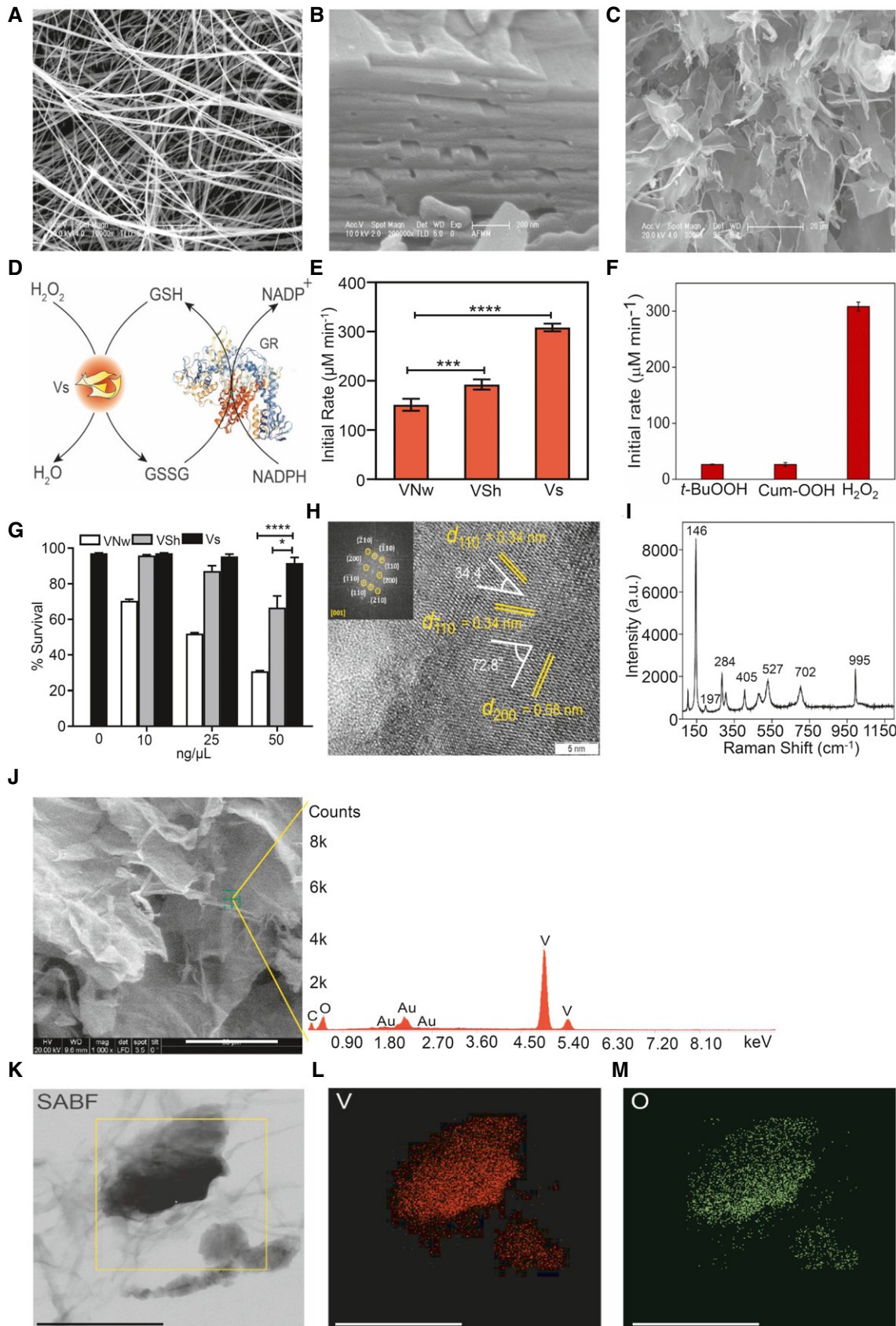

**Figure 1.**

◀

**Figure 1. Synthesis and characterization of vanadia (V₂O₅) nanoparticles.**

A–C   Scanning electron microscopy (SEM) images of (A) nanowires (VNw; scale – 5 μm) and (B) crude nanosheets (VSh; scale – 200 nm), (C) ultrathin nanosheets (Vs),
      scale – 20 μm.
D      Schematic representing the glutathione reductase (GR)-coupled assay to measure the GPX-like activity of Vs.
E      Comparison of initial activity rate among all three forms of V₂O₅ nanomaterials having a common exposed facet [001].
F      Bar diagram of the initial rate of Vs with 3 different peroxides, *t*-BuOOH—tertiary-butyl hydroperoxide, Cum-OOH—cumene hydroperoxide and H₂O₂.
G      U1 cells were treated with increasing concentrations of VNw, VSh, or Vs – 10 to 50 ng/μl- for 15 min, and cell survival was analyzed by flow cytometry after 24 h
      by propidium iodide (PI) staining.
H      High-resolution TEM (HRTEM) and fast Fourier transform (FFT) (inset) of Vs showing the lattice fringes and the exposed plane.
I       FT-Raman spectroscopy of Vs showing the peaks corresponding to the orthorhombic phase of the material.
J–M   Energy dispersive spectroscopy (EDS) of Vs (J). The small peak at 2.0 KeV is due to Au sputtering while recording the spectra (Scale – 50 μm). The peak of C is
      coming from atmospheric carbon. X-Ray mapping images of Vs (Scale – 300 nm). (K) Left column: Selective area bright field (SABF) image, (L) middle column:
      distribution of vanadium (V) atoms in red, (M) right column: distribution of oxygen (O) atoms in green.

Data information: *$P < 0.05$, ***$P < 0.001$, ****$P < 0.0001$ by Student's *t* test. (E) and (F) Data are representative of three independent experiment (mean ± SEM). (G)
Data are representative of two independent experiments done in duplicate (mean ± SEM).

## Vs displays efficient H₂O₂ linked GPX activity

Having established the biophysical characteristics of Vs, we examined the GPX-mimetic activity of Vs using GR-coupled assay. Various control experiments, such as reactions lacking GSH/GR/H₂O₂, were performed to rule out the possibility of nonspecific reactions. We observed that in the absence of any one of the required constituents, Vs does not show GPX-like activity, i.e., no reduction of H₂O₂ takes place (Fig 2A, and Appendix Fig S3A). Varying the concentration of Vs from 0 to 20 ng/μl led to a proportional dependence of the initial rate for the reduction of H₂O₂ with first-order reaction kinetics (Fig 2B). Since both H₂O₂ and GSH are important for the GPX activity, we performed the activity assay by varying concentrations of H₂O₂ (0–400 μM) and GSH (0–7 mM) under steady-state condition. Typical enzymatic Michaelis–Menten kinetics was observed for both H₂O₂ and GSH (Fig 2C and D). The corresponding Lineweaver–Burk plots are depicted in Fig 2E and F. For comparison, we simultaneously performed kinetics of Vs, VSh, and VNw. Interestingly, the $V_{max}$ values for Vs and VSh correlate with their surface area with the exception of VNw (Fig 2G and H). This is consistent with the differences in the {001} exposed facets among three morphologies. For H₂O₂, the $K_M$ values obtained for VNw, VSh, and Vs were 44.4 ± 1.7, 57.3 ± 3.8, and 112.2 ± 3.8 μM, respectively (Fig 2H). This indicates that the surface of the nanowires and nanosheets (VNw & VSh) are saturated at lower concentrations of H₂O₂ (Ghosh *et al*, 2018), whereas relatively higher concentrations of H₂O₂ are required for the saturation of the surface of ultrathin nanosheets (Vs).

The stability of the nanomaterials for the reduction of H₂O₂ was examined by performing multiple assay cycles, which demonstrate only a marginal loss of catalytic activity (Fig EV2A). TEM measurements of nanomaterial surface before and after multiple rounds of catalysis indicate no alterations (Fig EV2B–D), confirming that Vs performs H₂O₂ reduction with unprecedented stability, specificity, and sensitivity *in vitro*.

## Vs mimics GPX activity inside the HIV-1 infected cells

To test the Vs-related GPX activity inside mammalian cells and to understand its influence on HIV-1, we selected the U1 cell line model of HIV-1 latency and reactivation. The U1 cell line is derived from the parent promonocytic cell line U937, wherein two copies of

the HIV-1 genome are latently integrated (Folks *et al*, 1987). The viral replication can be induced by treatment of U1 cells with various pro-inflammatory agents such as phorbol myristate acetate (PMA), tumor necrosis factor-alpha (TNF-α), and granulocyte-macrophage colony-stimulating factor GM-CSF (Folks *et al*, 1987; Poli *et al*, 1990). We first examined the uptake of Vs nanomaterial by U1 through inductively coupled plasma mass spectrometry (ICP-MS). The U1 cells were treated with 50 ng/μl of Vs for 15 min, followed by extensive washing to remove Vs associated with the cell surface. The cells were then lysed, and the lysate was subjected to ICP-MS to examine Vs internalization. As shown in Fig 3A, ICP-MS of cell lysate showed a buildup of Vs inside U1, followed by gradual decrease over time such that only a fraction of internalized Vs was retained (Fig 3A). We further confirmed Vs internalization by U1 using SEM. The surface of Vs-treated U1 cells exhibited depressions as compared to untreated cells (Fig EV3A–C). To confirm that the observed depressions were due to uptake of Vs, we coupled SEM with electron dispersive X-ray spectroscopic (EDS) analysis of the concavity. The peak corresponding to vanadium was specifically detected in Vs-treated U1 (Fig EV3A–C). These observations are consistent with our earlier study showing efficient internalization of V₂O₅ nanowires by the mammalian cells (Vernekar *et al*, 2014).

To examine the role of Vs in the intracellular reduction of H₂O₂ through GPX activity, we exploited Orp1-roGFP2 and Grx1-roGFP2 biosensors that allow non-invasive imaging of the intracellular H₂O₂ and $E_{GSH}$, respectively (Gutscher *et al*, 2008; Gutscher *et al*, 2009; Morgan *et al*, 2011). The roGFP2 moiety has two surface exposed cysteines, which undergo disulfide bond formation upon oxidation resulting in an increase in fluorescence excitation intensity at 405 nm along with a relative decrease at 488 nm excitation at a fixed emission of 510 nm (Gutscher *et al*, 2008). An inverse relation in 405/488-biosensor ratio was detected upon the reduction of disulfides. The specific equilibration of the roGFP2 dithiol-disulfide redox pair (roGFP2_red/roGFP2_oxi) either with H₂O₂ or with glutathione redox pair (GSH/GSSG) is efficiently catalyzed by the covalently fused peroxidase Orp1 or glutaredoxin Grx1, respectively (Gutscher *et al*, 2009; Morgan *et al*, 2011). The fusion of Orp1 with roGFP2 creates a redox relay in which Orp1 mediates near-stoichiometric oxidation of roGFP2 by H₂O₂ (Gutscher *et al*, 2009) (Fig 3B). Likewise, Grx1 mediates oxidation of roGFP2 in response to a nanomolar increase in intracellular GSSG upon H₂O₂ stress (Gutscher *et al*, 2008). The redox relay created by Orp1-roGFP2 or Grx1-roGFP2

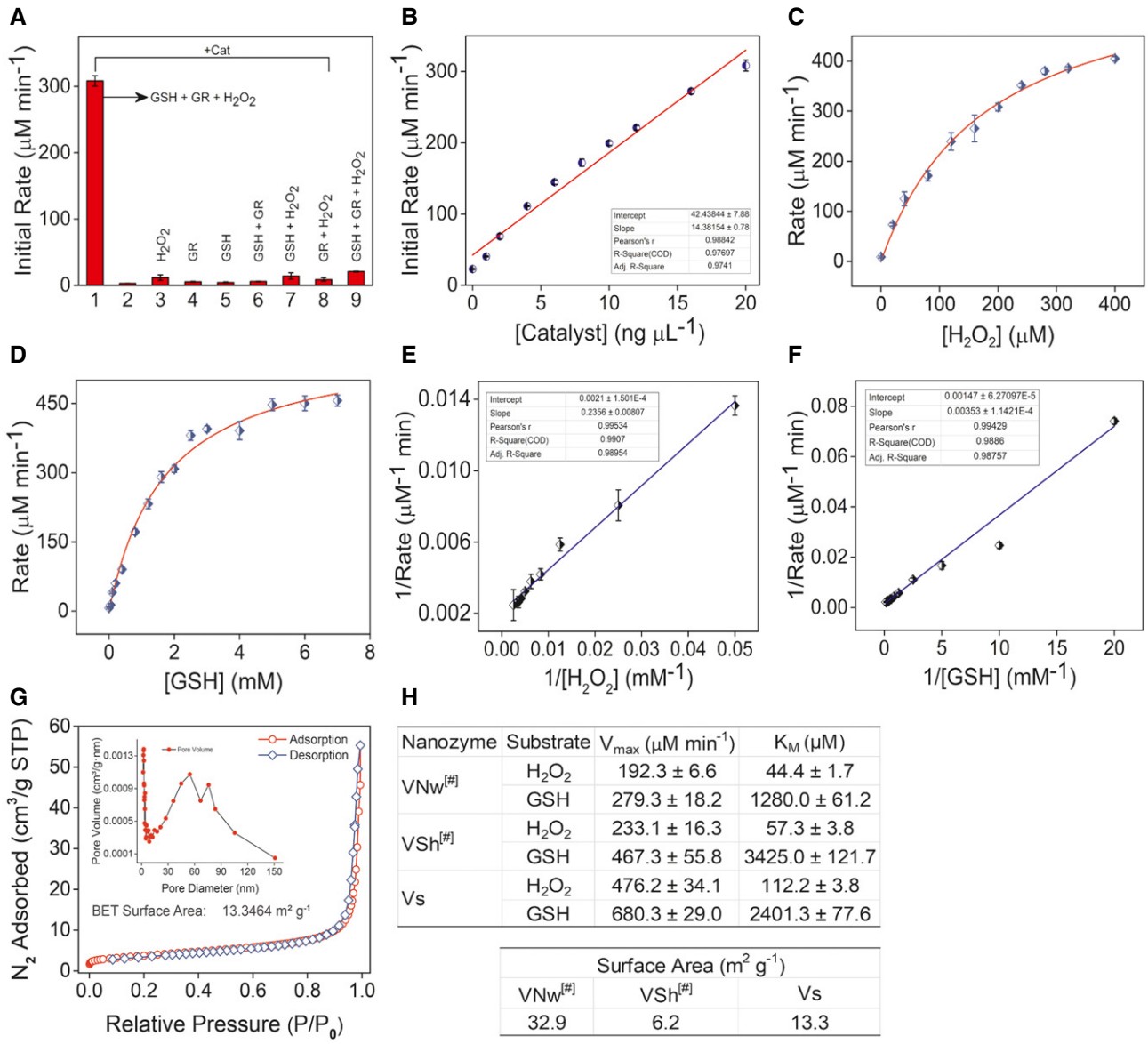

**Figure 2. Enzymatic parameters and surface area measurements of Vs.**

A   Bar diagram shows the initial rate of GPX-like activity of Vs (Cat – 20 ng/µl) under different assay conditions.

B   Dependence of the initial reaction rate for $H_2O_2$ reduction on varying concentration of the catalyst, Vs.

C   Michaelis–Menten plot with the variation of $H_2O_2$ (0–400 µM) in the presence of Vs (20 ng/µl), GSH (2 mM), NADPH (0.2 mM), GR (1.7 units) in phosphate buffer (100 mM, pH 7.4) at 25°C.

D   Michaelis–Menten plot with the variation of GSH (0–7 mM) in the presence of Vs (20 ng/µl), $H_2O_2$ (200 µM), NADPH (0.2 mM), GR (1.7 units) in phosphate buffer (100 mM, pH 7.4) at 25°C.

E, F  Lineweaver–Burk plot with varying concentration of $H_2O_2$ (E) and GSH (F) in presence of Vs nanozyme, respectively. The concentration of NADPH was constant (0.2 mM) in all the assay conditions.

G   Surface area measurement by $N_2$ adsorption, desorption isotherm, and distribution of pore size (Inset). The measured surface area of Vs was 13.3 m²/g.

H   Enzyme kinetic parameters and BET surface area values of different forms of $V_2O_5$ nanomaterials [#] reported from our previous literature (Ghosh et al, 2018).

Data information: Data are representative of three independent experiments (mean ± SEM).

demonstrates dynamic behavior as the biosensor ratio returns to basal during recovery from oxidative stress due to normalization of $H_2O_2$ and GSSG levels (Gutscher et al, 2008; Gutscher et al, 2009).

We created stable transfected U1 cells that express either Orp1-roGFP2 (U1-Orp1-roGFP2) or Grx1-roGFP2 (U1-Grx1-roGFP2) in the

cytosol (Appendix Fig S4A and B). Exposure of U1-Orp1-roGFP2 to $H_2O_2$ for 2 min showed a concentration-dependent increase in the biosensor ratio, consistent with the Orp1-mediated oxidation of roGFP2 by $H_2O_2$ (Fig 3C). In contrast, pretreatment of U1-Orp1-roGFP2 with 25 and 50 ng/µl of Vs for 15 min diminished biosensor

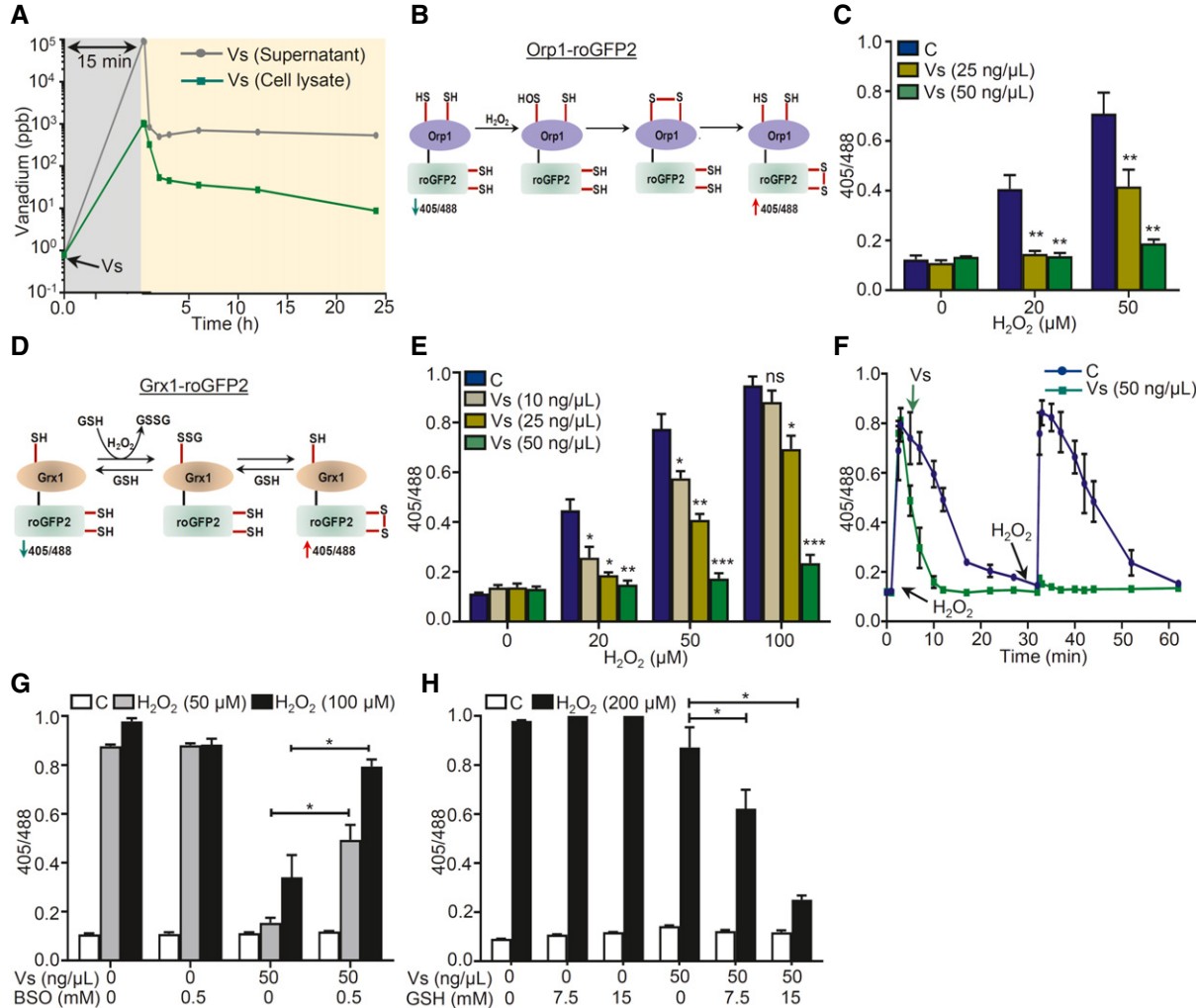

**Figure 3. Vs acts as a mimic of GPX in U1 cells.**

A    U1 cells were treated with 50 ng/µl of Vs for 15 min (gray area), washed, and lysed, and kinetics of Vs internalization and retention (yellow area) was quantified by subjecting cell lysate and supernatant to ICP-MS for measuring intracellular and extracellular Vs, respectively.

B    Schematic representation showing working principle of Orp1-roGFP2 biosensor.

C    U1-Orp1-roGFP2 cells were treated with Vs for 15 min, followed by exposure to $H_2O_2$ for 2 min, and ratiometric response was measured.

D    Schematic representation showing working principle of Grx1-roGFP2 biosensor.

E    U1-Grx1-roGFP2 cells were pre-treated with Vs for 15 min, exposed to $H_2O_2$ for 2 min, and ratiometric response was measured.

F    U1-Grx1-roGFP2 cells were treated with two doses of 50 µM $H_2O_2$ (black arrows) and the ratiometric response was measured (blue line). Parallelly, U1 cells treated similarly with $H_2O_2$ were exposed to Vs at the indicated time point and the ratiometric response was measured (green line). ****$P < 0.0001$, by Wilcoxon matched-pairs signed rank test.

G, H    U1 Grx1-roGFP2 cells were supplemented with BSO (G) or GSH (H) for 16 h to deplete or replenish GSH, respectively. Following this, cells were treated with Vs for 15 min and exposed to $H_2O_2$, and the ratiometric response was measured.

Data information: Data are representative of results from three independent experiments performed in duplicate (mean ± SEM). ***$P < 0.001$, **$P < 0.01$, *$P < 0.05$, by Mann–Whitney test. Asterisks (*) compare Vs-treated cells with control cells (C).

oxidation by $H_2O_2$, consistent with the Vs-catalyzed reduction of $H_2O_2$ (Fig 3C). Because $H_2O_2$ exposure also leads to oxidation of reduced GSH to GSSG (Perl & Banki, 2000), we monitored this transformation using Grx1-roGFP2 biosensor (Fig 3D). The U1-Grx1-roGFP2 cells were challenged with various concentrations of $H_2O_2$ for 2 min, and the sensor response was quantified. We found that the biosensor responds to increasing concentrations of $H_2O_2$ and treatment with 100 µM of $H_2O_2$ for 2 min results in 90% oxidation

of Grx1-roGFP2 (Fig 3E). The corresponding $E_{GSH}$ was −240 mV, which is higher than the basal $E_{GSH}$ for U1 cells, −320 mV. Pretreatment of U1-Grx1-roGFP2 with Vs for 15 min effectively reduced $H_2O_2$-mediated oxidation of biosensor in a concentration-dependent manner (Fig 3E). Next, we measured the time kinetics of Grx1-roGFP2 oxidation to a low concentration of $H_2O_2$ (50 µM). An increase in the biosensor ratio was observed within 2 min of $H_2O_2$ exposure followed by a gradual decrease to the baseline levels in

30 min, indicating efficient mobilization of cellular antioxidant machinery (Gutscher et al, 2008; Fig 3F). In contrast, the addition of 50 ng/µl of Vs at post $H_2O_2$ treatment decreased the biosensor oxidation to baseline levels within 10 min (Fig 3F). Importantly, a single dose of Vs completely prevented subsequent oxidation of biosensor by $H_2O_2$ (Fig 3F). These data are fully consistent with earlier results demonstrating multiple cycles of $H_2O_2$ reduction by a single dose of Vs in vitro.

Since GPX function is dependent on GSH as an electron donor (Brigelius-Flohe & Maiorino, 2013), we tested the requirement of GSH in Vs-mediated $H_2O_2$ reduction. We treated U1-Grx1-roGFP2 cells with 0.5 mM buthionine sulfoximine (BSO), which lowers cellular GSH content by inhibiting γ-glutamylcysteine synthetase (GCS) activity (Drew & Miners, 1984). Following this, cells were treated with Vs for 15 min and exposed to 50 and 100 µM of $H_2O_2$ for 2 min. As shown earlier, both the concentrations of $H_2O_2$ achieved nearly complete oxidation of Grx1-roGFP2, which was effectively blocked by Vs pretreatment (Fig 3G). In contrast, pretreatment with BSO attenuated Vs ability to prevent biosensor oxidation by $H_2O_2$ (Fig 3G). Supplementation of exogenous GSH (15 mM) restored Vs activity as shown by a significant decrease in the biosensor oxidation upon challenge with a saturating concentration of $H_2O_2$ (Fig 3H). Lastly, we also examined if Vs activity is influenced by the Trx/TrxR antioxidant system. For this, we pretreated U1-Grx1-roGFP2 cells with various concentrations of the TrxR inhibitor, auranofin and examined the ability of Vs to protect biosensor oxidation by $H_2O_2$. As shown in the Fig EV4A, auranofin did not affect the ability of Vs to reduce $H_2O_2$, confirming the GSH-dependent GPX activity of Vs.

## Vs subverts HIV-1 reactivation

Studies have shown that $H_2O_2$ treatment reactivates HIV-1 from latency (Legrand-Poels et al, 1990; Bhaskar et al, 2015). Increased oxidative stress was shown to activate the HIV-1 LTR through redox-sensitive transcription factors, such as NF-κB (Pyo et al, 2008). On this basis, we reasoned that Vs displaying efficient antioxidant activity could affect redox-dependent reactivation of HIV-1. We first induced HIV-1 expression using low concentrations of PMA (5 ng/ml) and prostratin (1.25 µM), two well-established activators of HIV-1 (Kim et al, 1996; Gulakowski et al, 1997). The expression of the HIV-1 gag transcript was monitored as a marker of HIV-1 activation by RT–qPCR at various time points post-treatment with PMA/prostratin. Both activators induced HIV-1 transcription with a significant increase observed at 24 h post-treatment (Fig 4A). Pre-exposure of U1 with Vs or N-acetyl cysteine (NAC- a well-established antioxidant) effectively blocked PMA/prostratin-mediated viral reactivation (Fig 4A and B). Using U1-Orp1-roGFP2, we confirmed an increase in the intracellular levels of $H_2O_2$ at 6 and 12 h post-PMA treatment, which was significantly reduced upon Vs pretreatment (Fig 4C). This indicates that oxidative stress precedes PMA-stimulated virus reactivation and GPX activity associated with Vs counteracted redox-dependent HIV-1 reactivation. The capacity of Vs in lessening HIV-1 activation was also confirmed in a lymphocytic model of HIV-1 latency (J1.1) (Fig 4D), corroborating that the effect of Vs is not restricted to a cell type.

HIV-infected individuals suffer from selenium (Se) deficiency that adversely affects the activity of Se-dependent GPX enzyme leading to oxidative stress, HIV reactivation, and exacerbation of disease pathology (Look et al, 1997a; Campa et al, 1999). Therefore, Se limitation is a physiologically relevant stimulus that induces oxidative stress and HIV-1 reactivation (Look et al, 1997b). We envisage that Se-independent GPX activity of Vs could replenish the impaired activity of cellular GPX under Se-deficient conditions to subvert HIV-1 reactivation. To examine this, we starved U1-Grx1-roGFP2 of fetal bovine serum (FBS; the source of Se) and monitored the change in its antioxidant response over time. We observed an increase in biosensor ratio within 30 min of FBS removal, indicating oxidative stress (Appendix Fig 5A). Supplementation of Vs or Se in the culture medium of Se-deficient U1-Grx1-roGFP2 decreased biosensor ratio, signifying alleviation of oxidative stress by Vs (Fig 4E). As expected, Se-deficiency triggered HIV-1 reactivation in U1, and addition of Vs or Se had an opposite effect (Fig 4F).

## Vs dampens the expression of host genes involved in HIV-1 reactivation

Having shown the utility of Vs in countering oxidative stress and HIV-1 reactivation, we next examined the underlying mechanisms. We performed expression analysis using the NanoString nCounter system, which permits absolute quantification of multiple RNA transcripts without any requirements for reverse transcription (Kulkarni, 2011). We focused on 185 host genes that are known to respond to HIV infection and oxidative stress (Appendix Table S1–S3). We performed expression analysis on RNA isolated from U1, PMA-treated U1, Vs-treated U1, and Vs plus PMA-treated U1. The fold change (> 1.5-fold, $P < 0.05$) was calculated by normalizing the raw mRNA counts to the geometric mean of the internal control $\beta_2$ microglobulin (B2M).

A total of 118 genes showed differential expression under the conditions tested (Fig 5A). Overlap analysis confirmed 55 of 118 genes to be common in each category (Fig 5B). Treatment with PMA induced the expression of genes associated with ROS and RNS (reactive nitrogen species) generation (e.g., NADPH oxidase subunits [NCF1, NCF2] and nitric oxide synthase [NOS2]). Genes involved in antioxidant response, including catalase (CAT) and superoxide dismutase 2 (SOD2), were down-regulated upon PMA treatment (Fig 5C). Overall, these changes are consistent with increased oxidative stress in response to PMA triggered HIV-1 reactivation (Bhaskar et al, 2015). Up-regulation of genes involved in reducing free iron pool (ferritin heavy chain 1; FTH1), maintaining GSH balance (cysteine/glutamate transporter; SLC7A11), (Fig 5C), indicate a compensatory mechanism to protect from oxidative conditions induced by PMA (Sato et al, 2005; Eid et al, 2016). The transcription factor Nrf2 is the major activator of antioxidant systems (Espinosa-Diez et al, 2015). Surprisingly, a majority of Nrf2 dependent antioxidant systems such as GSH biosynthesis/recycling (e.g., GSS, GPX1, GPX4, GSTP1), thioredoxins (e.g., TXNRD2), and peroxiredoxins (PRDX6) were down-regulated upon treatment with Vs alone or Vs plus PMA (Fig 5C), indicating an adverse influence of Vs on U1 cells. One likely possibility is that the natural antioxidant defense mechanisms are attenuated by a feedback-like mechanism because of the potent antioxidant properties of Vs. Also, the expression of superoxide producing system (NCF1 and NCF2) was repressed in Vs alone and Vs plus PMA-treated U1 (Fig 5C), which can further reduce intracellular ROS levels.

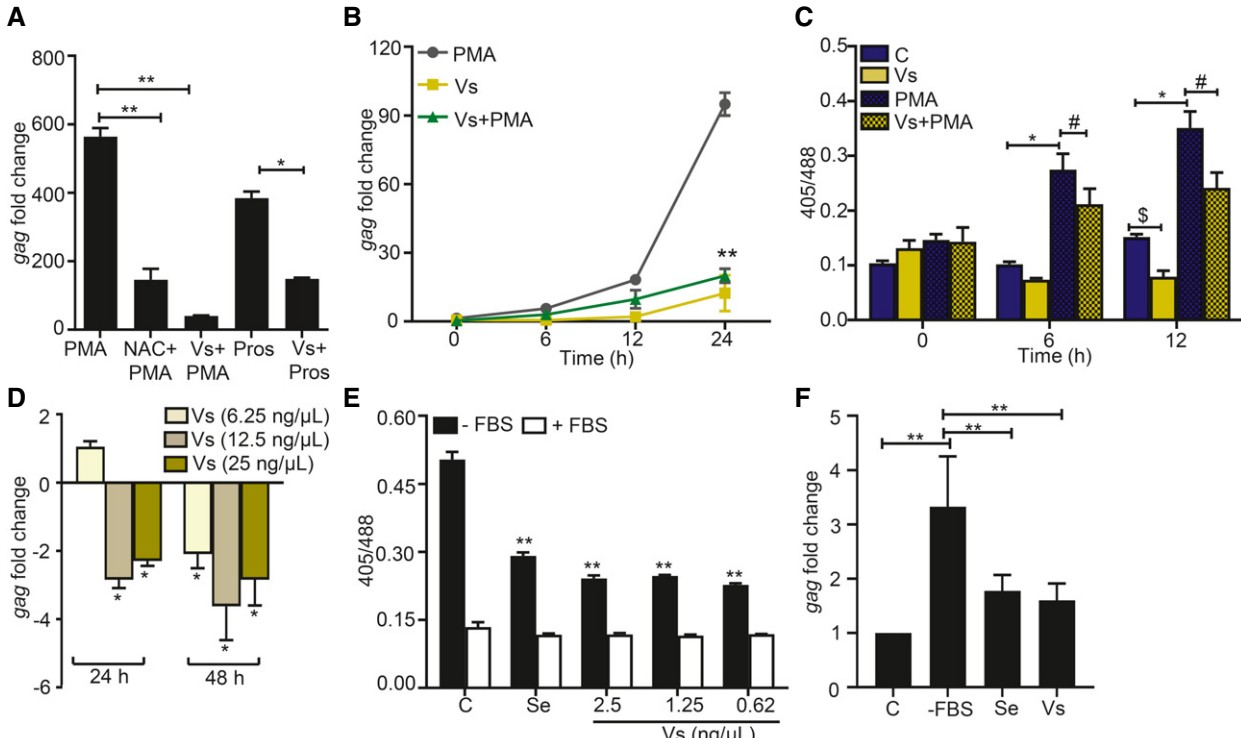

**Figure 4. Vs subverts HIV reactivation in U1 cells.**

A  Vs-treated U1 cells were challenged with 5 ng/ml PMA or 1.25 μM prostratin (Pros) for 24 h, and HIV-1 induction was monitored by *gag* RT–PCR. 10 mM NAC, an antioxidant known to subvert PMA mediated viral reactivation, was used as a positive control.

B  Vs-treated U1 cells were exposed to PMA, and viral activation was measured as a function of time, by *gag* RT–PCR. U1 cells were also treated with Vs or PMA alone.

C  Untreated or Vs-treated U1-Orp1-roGFP2 cells were exposed to PMA and the biosensor response was measured at the indicated time points. The biosensor response was also measured for untreated or PMA-treated cells.

D  J1.1 cells were treated twice with Vs for 15 min at 0 and 24 h time point. HIV-1 induction was measured by *gag* RT–PCR at 24 h and 48 h post-treatment. An untreated control was used for normalization.

E  U1-Grx1-roGFP2 cells were serum starved for 30 min in the presence or absence of Vs and sodium selenite (0.5 nM), and the biosensor response was measured. Data were compared to serum-starved control cells (C).

F  U1 cells were either serum-starved or supplemented with Se (0.5 nM) or Vs (0.62 ng/μl), and HIV reactivation was measured at 6 h post-starvation by *gag* RT–PCR.

Data information: \*\*$P < 0.01$, $^{\$/\#/*}P < 0.05$, by Mann–Whitney test. Data are representative of results from three independent experiments performed in triplicate (mean $\pm$ SD).

Genes known to be associated with HIV-1 activation such as transcription factors (*e.g.,* FOS and CEBPB) (Roebuck *et al,* 1996; Henderson & Calame, 1997), inflammatory cytokines/receptors (TGFβ1, TNFRSF1B, and IL16; Hu *et al,* 1996; Herbein *et al,* 1998; Amiel *et al,* 1999), and chemokines (CCL3 and CCL4) (Choe *et al,* 2001) were induced upon PMA treatment and repressed by Vs plus PMA (Fig 5C). Several genes encoding proteins associated with HIV-1 replication, packaging, budding, and fitness (*e.g.,* APOBEC3G, CD44, XPO1, VPS4A, DHCR24) were down-regulated upon Vs plus PMA treatment as compared to PMA alone (Fig 5D). It is known that cells latently infected with HIV-1 are refractory to apoptosis, whereas increased apoptosis promotes HIV-1 reactivation (Khan *et al,* 2015). Consistent with this, a majority of genes encoding pro-apoptotic proteins (*e.g.,* BAD, BAX, CASP3, and CASP8) were substantially repressed upon Vs plus PMA treatment as compared to PMA alone (Fig 5C). In addition, a cellular inhibitor of transcription factor NF-κB (i.e., NFKBIA) was highly induced upon Vs or Vs plus PMA treatment (Fig 5D). Since NF-κB is critical for HIV-1 reactivation (Staal *et al,* 1990), the increased expression of its inhibitor (NFKBIA) by Vs

is indicative of attenuated reactivation of HIV-1. Based on this, we hypothesize that a well-established pharmacological inhibitor of NF-κB (*E*)3-[(4-methylphenyl)sulfonyl]-2-propenenitrile (BAY11-7082) (Devadas *et al,* 2004) would synergize with Vs to efficiently subvert HIV-1 reactivation. To examine this, we exposed U1 cells pre-treated with Vs to BAY11-7082 and HIV-1 reactivation in response to PMA was monitored by measuring the levels of *gag* transcript and p24 capsid protein. Consistent with our hypothesis, exposure of U1 to both Vs and BAY11-7082 suppresses reactivation of HIV-1 which supersedes that produced by either Vs or BAY11-7082 alone (Fig 5E and F). Overall, Vs not only affected the expression of redox pathways but as a consequence also modulates the expression of pathways coordinating the inflammatory response, viral fitness, transcription, and apoptosis to subvert HIV-1 reactivation.

## Vs adversely affects intracellular replication of HIV-1

Along with reactivation, oxidative stress is associated with replication of HIV-1 in cell lines and primary CD4$^+$ T-cell and macrophages

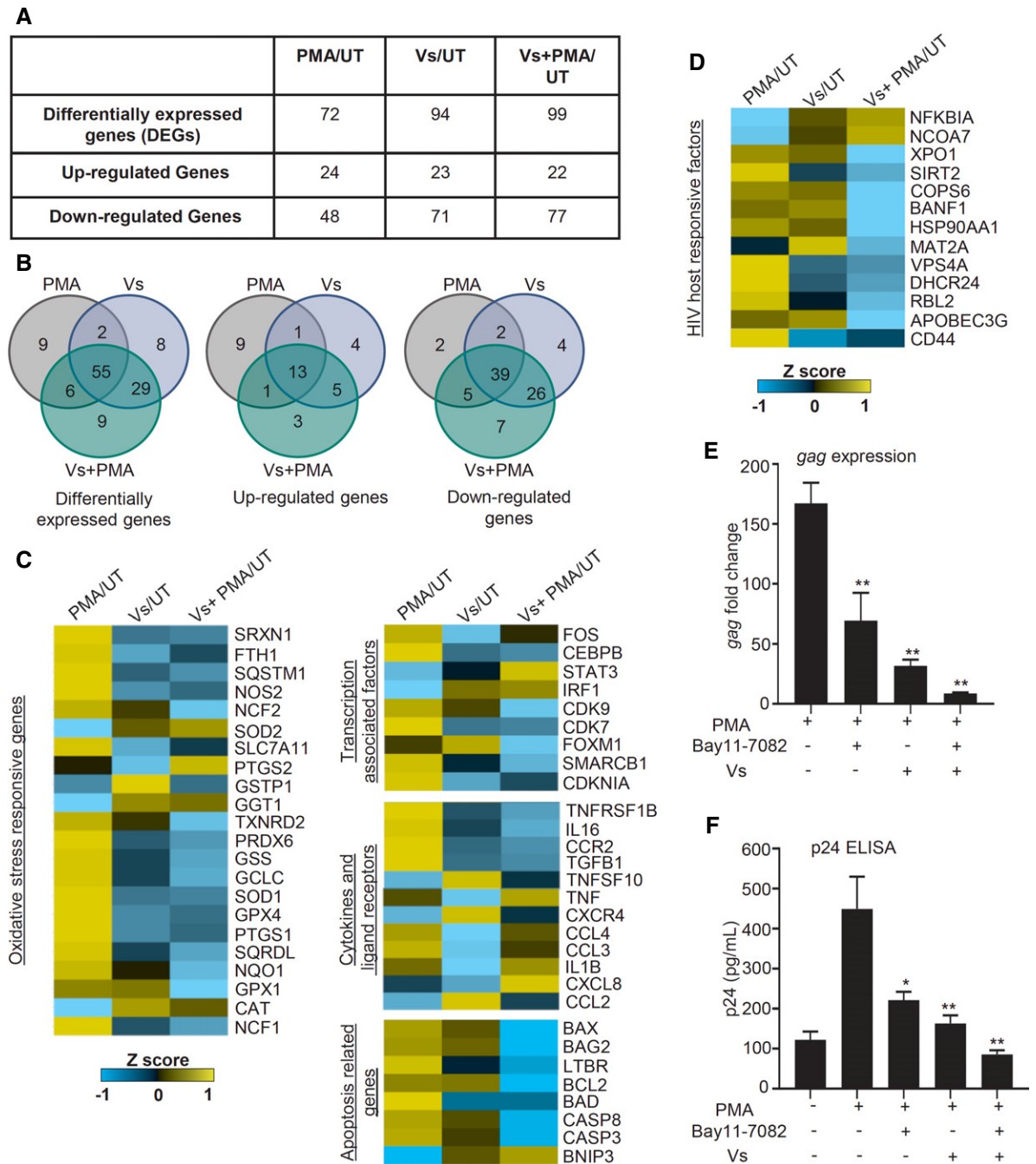

**Figure 5.** Vs modulates the expression of genes mediating oxidative stress response, HIV-1 activation, inflammation, and apoptosis.

A–D Total RNA isolated from untreated (UT), PMA-treated, Vs-treated, and Vs + PMA-treated U1 was examined by NanoString technology to assess the expression of genes responsive to oxidative stress and HIV. (A) Differentially expressed genes (DEGs) under the indicated conditions. (B) The Venn diagram of DEGs significantly perturbed under different comparison conditions. (C and D) Heat map showing functional categories of DEGs under PMA/UT, Vs/UT, and PMA + Vs/UT comparisons. mRNA counts were normalized using the internal control $\beta_2$ microglobulin (B2M), and fold change (FC) was calculated using the nSolver 4.0 software. Genes showing an absolute FC > 1.5, and $P < 0.05$ were considered as significantly altered.

E, F Vs-treated or untreated U1 cells were exposed to Bay11-7082 (7.5 μM). PMA mediated HIV reactivation at 12 h was monitored by *gag* RT–PCR and p24 ELISA.

Data information: Data are representative of results from three independent experiments performed in triplicate (mean ± SEM). **$P < 0.01$, *$P < 0.05$, by Mann–Whitney test. Asterisks (*) compare different treatment conditions with PMA-treated cells.

(Perl & Banki, 2000; Aquaro *et al*, 2007; Bhaskar *et al*, 2015; Tasker *et al*, 2016; Shytaj *et al*, 2020). Therefore, we next examined the influence of Vs activity on HIV-1 replication. First, we used a stable

CD4[+] T-cell line expressing EGFP (CEM-GFP) under HIV-1 LTR. Infection with HIV-1 significantly induces the expression of CEM-GFP (Gervaix *et al*, 1997). The infection of CEM-GFP cells with

CXCR4-using virus, (HIV-1 NL-4.3) progressively increased GFP fluorescence over 5 days (Fig 6A). Addition of 50 ng/µl of Vs for 15 min every 24 h completely blocked GFP expression in the infected CEM-GFP cells (Fig 6A). Exposure to only a single dose of Vs for 15 min did not affect GFP expression, whereas 15 min of Vs exposure every 48 h partially reduced expression (Appendix Fig 6A–C). These results indicate that pretreatment for 15 min every 24 h is required to block HIV-1 replication by Vs and the protective effect is likely to be reversible. We also infected Jurkat CD4$^+$ T cells with HIV-1 NL-4.3 and measured *gag* transcript, and p24 HIV capsid protein in the whole cell lysate and in the supernatant. Each technique showed a time-dependent increase in HIV-1 replication, which was efficiently inhibited by Vs (Fig 6B–D). Finally, we infected U937 promonocytic cells with CCR5-using virus, HIV-1 NL-AD8, and viral replication was estimated by measuring *gag* transcript at 24 h post-treatment. As shown in Fig 6E, HIV-1-infected U937 showed a 15-fold increase in *gag* transcript, which was reduced to 3-fold in case of Vs pretreatment (Fig 6E).

We also examined whether the antioxidant potential of Vs confers antiviral response in primary human CD4$^+$ T lymphocytes. We pre-treated primary CD4$^+$ T cells isolated from peripheral blood mononuclear cells (PBMCs) of three human donors with Vs (25 ng/µl), infected with HIV-1 NL4.3, and measured p24 HIV capsid protein in the supernatant at 3- and 5-day post-infection. The p24 ELISA confirmed a time-dependent increase in virus load, which was uniformly reduced upon pretreatment of primary CD4$^+$ T cells with Vs (Fig 6F). The reduction in viral load by Vs pretreatment was also confirmed in human monocyte-derived macrophages (HMDM) infected with HIV-1 AD8 (Fig EV5A). Since GSH and Trx/TrxR systems protect HIV-infected cells from apoptosis (Chirullo *et al*, 2013; Bhaskar *et al*, 2015), we assessed if the antioxidant potential of Vs influence apoptosis in primary CD4$^+$ T cells infected with HIV-1. For this purpose, we measured the frequency of Annexin V$^+$, propidium iodide (PI)$^+$, and Annexin V$^+$/PI$^+$ cells as predictive markers for early apoptosis, necrosis, and late apoptosis, respectively. The fraction of Annexin V$^+$ or PI$^+$ cells was not affected upon Vs pretreatment. However, the frequency of cells exhibiting late apoptosis (Annexin V$^+$/PI$^+$) was significantly reduced upon Vs pretreatment (Fig EV5B). These results are consistent with our data showing induction of anti-apoptotic genes by Vs and the cytoprotective potential reported for V$_2$O$_5$-based antioxidant nanozymes (Vernekar *et al*, 2014). In sum, Vs antioxidant activity efficiently counteracts the replication of HIV-1.

### Vs blocks viral reactivation in CD4$^+$ T cells isolated from virally suppressed patients

Next, we sought to examine whether Vs could prevent viral reactivation upon stimulation of latently infected CD4$^+$ T cells isolated from the PBMCs of three HIV-infected subjects on suppressive antiretrovirals (ARVs) for a minimum of six years as described (Kessing *et al*, 2017). The CD4$^+$ T cells were initially expanded in the presence of interleukin-2 (IL-2), phytohemagglutinin (PHA), "feeder cells", or with antiretrovirals (ARVs -efavirenz, zidovudine, and raltegravir) alone or ARVs together with Vs (Vs + ARVs) (Fig 7A). From day 7 onwards, CD4$^+$ T cells were cultured in a medium containing IL-2 and ARVs alone or Vs + ARVs (Fig 7B). The viral RNA increased initially, followed by low to undetectable

levels by day 21 (Fig 7B). Interestingly, viral RNA levels from all three subjects' cells exposed to Vs + ARVs were below detection compared to only one subject's cells in the case of ARVs alone (Fig 7B). Since Vs down-regulated the expression of Nrf2 dependent antioxidant systems in U1, we performed RT–qPCR analysis of a selected set of Nrf2-dependent genes (GPX1, GPX4, GSR, and SRXN1) in CD4$^+$ T cells at 21 days post-treatment with ARVs alone or Vs + ARVs. Consistent with findings in U1, expression of these genes was uniformly down-regulated in cells treated with Vs + ARVs compared to ARVs alone (Appendix Fig S7A).

We also assessed whether long-term treatment with Vs affected HIV reactivation. On day 21, we stimulated the CD4$^+$ T cells with the protein kinase C (PKC) activator prostratin in the absence of any treatment (Fig 7B). The activation of viral transcription was measured 24 h later by RT–qPCR. When ARVs were removed, followed by prostratin stimulation, the viral transcript was detected in all subjects' cells (Fig 7C and D). In contrast, upon Vs + ARVs removal followed by prostratin stimulation, viral transcription was inhibited by 90%, 100%, and 100% from three subjects, respectively, with an average inhibition of 96.7% for all three subjects in 24 h.

Lastly, we measured the total HIV-1 DNA to confirm that the reduction in viral reactivation by Vs is not due to the selective loss of latently infected cells. The total HIV-1 DNA content was comparable between the freshly isolated patients' CD4$^+$ T cells (*ex vivo*) and the expanded cells treated with Vs + ARVs or ARVs for the experiment's entire duration (Fig 7E). Also, using aqua dead cell stain, we confirmed that viability of expanded patients' CD4$^+$ T cells was not adversely affected upon prolong exposure to Vs + ARVs (Appendix Fig S8A). This suggests that we have not lost the cells with the capacity to reactivate the virus or selected for a fraction of cells that is non-responsive to prostratin. The inhibition of HIV reactivation is the consequence of Vs-mediated inhibition of the viral transcription without affecting proviral content. Altogether, the data show that Vs can potently inhibit viral reactivation in latently infected CD4$^+$ T cells isolated from ARV-suppressed HIV-1 infected individuals.

## Discussion

Studies exploring the application of antioxidant nanozymes in targeting human pathogens are limited. In-depth cellular studies using laboratory models that mimic the physiological environment during infection could help predict the clinical potential of nanozymes and will encourage new designs that are efficacious in humans. Using several biophysical, biochemical, cellular, genetic biosensors, and expression technologies, we demonstrated the biomedical application of V$_2$O$_5$-based nanozymes in counteracting redox stress and reactivation of HIV-1.

Until now, studies examining the antioxidant function of nanozymes relied on *in vitro* enzymatic assays or chemical analyses of redox metabolites (*e.g.*, GSH/GSSG) in whole cells or tissues. These invasive methods introduce oxidation artifacts and preclude observation of real-time changes in redox physiology upon nanozyme treatment. We circumvented these issues by applying non-invasive genetically encoded biosensors of H$_2$O$_2$ (Orp1-roGFP2) and $E_{GSH}$ (Grx1-roGFP2) to dynamically assess the activity of Vs in reducing

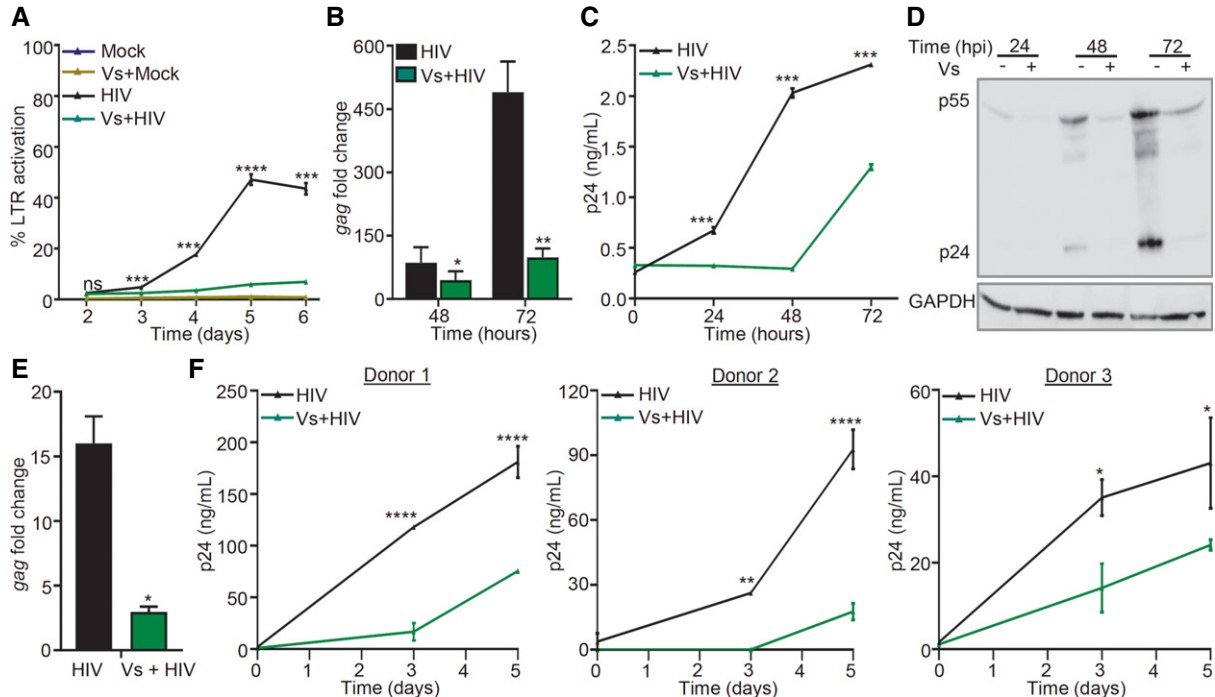

**Figure 6.  Vs reduces replication of HIV-1.**

A CEM-GFP cells were pre-treated with 50 ng/µl of Vs for 15 min and infected with 0.1 moi of CXCR4-using HIV-1 (NL-4.3), and GFP fluorescence was measured at 488 nm as an indicator of HIV LTR activity. Vs treatment was repeated every 24 h for the experiment.

B–D A similar assay was performed using Jurkat (CD4[+] T-cell line), and viral replication was assessed by (B) *gag* RT–PCR, (C) p24 ELISA in the culture supernatant, and (D) immunoblotting for p24 (viral capsid protein) in the whole cell lysate.

E U937 (promonocytes) were pre-treated with 50 ng/µl of Vs for 15 min followed by infection with 1 moi of CCR5 using HIV-1 (NL-AD8), and viral replication was measured by *gag* RT–qPCR at 24 h post-infection (hpi).

F Primary CD4[+] T cells purified from human PBMCs (3 healthy donors) were activated, pre-treated with 25 ng/µl Vs for 15 min, and infected with 0.05 moi of HIV-1 NL-4.3. Virus released in supernatant was quantified by p24 ELISA. Vs treatment was repeated every 48 h.

Data information: All figures except (B) and (E) were analyzed by 2-way ANOVA. (B), and (E) were analyzed by Mann–Whitney test. ****$P < 0.0001$, ***$P < 0.001$, **$P < 0.01$, *$P < 0.05$. Data are representative of results from three independent experiments performed in triplicate (mean ± SD).

Source data are available online for this figure.

intracellular $H_2O_2$ and maintaining GSH homeostasis. Numerous studies have indicated a link between GPX activity and HIV-1 *in vitro* and *in vivo* (Sappey *et al*, 1994; Look *et al*, 1997a). Remediation of $H_2O_2$ by GPXs potently reversed NF-κB-mediated HIV-1 transcription (Sappey *et al*, 1994). We have demonstrated that latently infected cells efficiently metabolized $H_2O_2$ likely via endogenous GPXs (Bhaskar *et al*, 2015). Expression of GPXs was elevated in monocytes and lymphocytes harboring latent HIV-1, whereas expression was diminished in cells and in PBMCs of patients during active HIV-1 replication (Bhaskar *et al*, 2015). Since the activity of cellular GPXs is selenium (Se)-dependent, studies have found a high correlation between Se-deficiency, GPX activation, and HIV-related mortality (Campa *et al*, 1999). Importantly, HIV-1-infected T cells showed the general downregulation of cellular Se proteins (Gladyshev *et al*, 1999). We found that Se depletion induced the oxidative shift in $E_{GSH}$ and promotes HIV reactivation. Importantly, Vs supplementation under Se depleted conditions was sufficient to diminish oxidative stress and HIV-1 reactivation. Mechanistically, our Nanostring data confirm that GPX-like activity of Vs efficiently suppresses the expression of redox-dependent transcription factors, pro-

inflammatory cytokines/chemokines, and pro-apoptotic molecules required for HIV-1 reactivation. We also found that Vs significantly reduced the fraction of human CD4[+] T cells exhibiting late apoptosis without affecting necrosis during HIV infection. Our findings are consistent with the requirement of efficient antioxidant potential to resist apoptosis, and death of chronically infected cells (Pinti *et al*, 2003; Fernandez Larrosa *et al*, 2008; Bhaskar *et al*, 2015). Agreeing to this, efficient inhibition of the Trx/TrxR antioxidant system by auranofin allowed targeting of HIV reservoirs by exerting pro-differentiating and pro-apoptotic effects (Chirullo *et al*, 2013). Interestingly, a combination of auranofin, BSO, and ART led to complete clearance of SIV viremia in macaques with 100% AIDS-free survival for at least 2 years after therapy interruption (Shytaj *et al*, 2013; Shytaj *et al*, 2015).

The sustained induction of Nrf2-driven cellular antioxidant response facilitates the successful transition between productive and latent HIV-1 infection (Bhaskar *et al*, 2015; Shytaj *et al*, 2020). Inhibition of Nrf2 promoted viral transcription and increased ROS generation (Shytaj *et al*, 2020), whereas its activation reduced HIV-1 infection (Furuya *et al*, 2016). Altogether, these studies indicate that

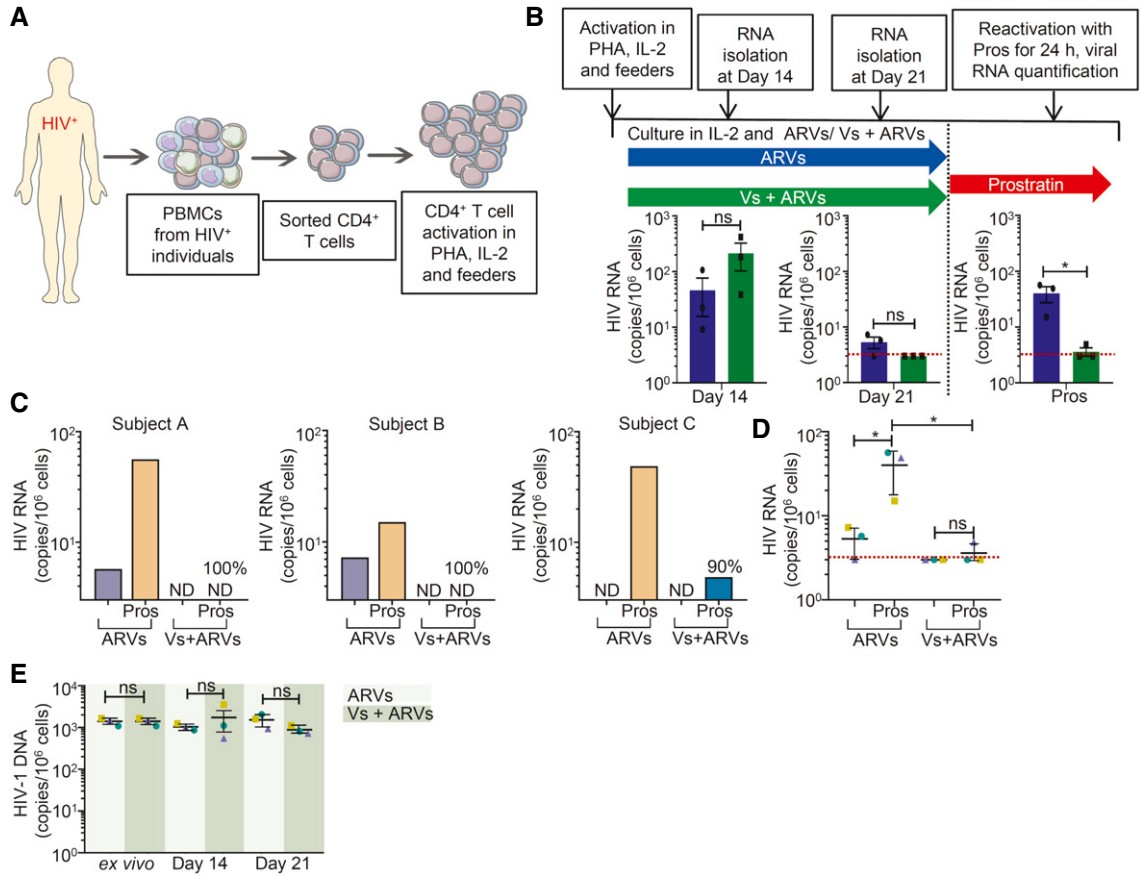

**Figure 7. Vs reduces HIV reactivation in primary CD4+ T cells isolated from virally suppressed individuals.**

A Schematic representation of generation of expanded CD4+ T cells and reactivation. CD4+ T cells were sorted from PBMCs of ARV-suppressed HIV-infected individuals and expanded in presence of PHA, IL-2, and autologous feeder PBMCs from healthy donor.

B Expanded CD4+ T cells from three patients were cultured in presence of IL-2 and ARVs, with and without 25 ng/μl Vs for 21 days. Vs treatment was given for 15 min every 3rd day. HIV transcripts were quantified by RT–qPCR at day 14, day 21, and at 24 h post-stimulation of cells cultured for 21 days by prostratin. Limit of detection for RT–qPCR was 3 viral transcripts per million cells.

C At day 21, cells were stimulated with 1 μM prostratin for 24 h and HIV transcripts were quantified by RT–qPCR. Reduction in viral stimulation in Vs-treated samples are represented as percentage values. ND—non-determined.

D Aggregate plot for 3 patients from data (C).

E Total HIV-1 DNA was determined up to 21 days in cells treated with ARVs or Vs + ARVs.

Data Information: (B), (D), and (E) were analyzed by one-way ANOVA with Tukey's multiple correction. *P < 0.05, ns—non-significant. Data are aggregated from three ARV-suppressed HIV-infected human subjects (mean ± SD).

---

diminished intracellular ROS levels contribute to latency induction and maintenance, whereas oxidative stress-inducing drugs likely reverse viral latency (Savarino *et al*, 2009; Yang *et al*, 2009). Our findings indicate that Vs treatment not only inhibited HIV reactivation and replication but also reduced the expression of Nrf2-dependent antioxidant genes. The HIV-1 proteins such as gp120, Nef, Tat, Vpr, and reverse transcriptase are known to elevate ROS production and induce Nrf2 pathway (Ivanov *et al*, 2016; Mastrantonio *et al*, 2016). Therefore, it is likely that anti-HIV properties of Vs would have reduced the levels of the HIV proteins necessary to generate oxidative stress. This, along with efficient GPX activity of Vs, possibly diminished the endogenous ROS levels significantly lower than that necessitates the mobilization of Nrf2-driven antioxidant expression.

Finally, along with several latently infected cell lines, we exploited primary cells derived from HIV-infected individuals and

confirmed that Vs impaired the ability of HIV to reactivate. Importantly, the use of expanded primary CD4+ T cells from HIV-1-infected patients allowed us to examine the effect of Vs on cells carrying autologous virus, thereby overcoming the limitation associated with the use of clonal HIV-1 laboratory strains (Jordan *et al*, 2003). Our findings raise the possibility of including Vs with the frontline treatment for faster suppression and potentially reducing the reservoir's size. It has been suggested that transient viral replication ("blips") observed in plasma could be reseeding the reservoir even in the presence of ART (Ramratnam *et al*, 2004; Jones & Perelson, 2007). Furthermore, therapy non-compliance can also result in viremia and reservoir replenishment. Combining Vs in ARV regimens could potentially inhibit reservoir replenishment during these situations. Lastly, clinically relevant ARVs have been shown to induce massive ROS (Mondal *et al*, 2004; Weiss *et al*, 2016), which

may contribute to the development of cardiovascular diseases and CNS pathologies (Hurwitz *et al*, 2004; Masia *et al*, 2007; Opii *et al*, 2007). The inclusion of antioxidant nanozymes such as Vs could help alleviate ARV-induced ROS production to improve therapy outcomes. Future experiments should be aimed to understand the underlying mechanism and how long Vs extends suppression of HIV transcription in combination with ARVs or even after treatment interruption.

# Materials and Methods

## Preparation of different morphologies of vanadium pentoxide (V$_2$O$_5$) nanoparticles

V$_2$O$_5$ nanowires (VNw) were synthesized by hydrothermal method described earlier (Ghosh *et al*, 2018). Briefly, 1.5 g of VOSO$_4$.H$_2$O and 0.835 g of KBrO$_3$ were sequentially dissolved in 30 ml ultrapure water and stirred for 45 min to form a yellow precipitate. The precipitate was dissolved in 4.5 ml concentrated HNO$_3$ with continuous mixing for 30 min and poured in a 50 ml teflon-lined stainless steel autoclave, heated to 190°C for 24 h in an oven, and then cooled down to room temperature (RT). The bright yellow color precipitate formed was filtered with 0.22 μm membrane filter paper and was washed thoroughly with double distilled water until the pH of the filtrate reached ~7.0. The precipitate was then washed twice with absolute ethanol and dried at 70°C for 12 h.

Ultrathin V$_2$O$_5$ nanosheets (Vs) were synthesized from the crude V$_2$O$_5$ nanosheets (VSh). Briefly, 2 mM of V$_2$O$_5$ powder was dispersed in 15 ml ultrapure water for 20 min. Then, 15 ml H$_2$O$_2$ (30% w/v) was added dropwise. During the addition of H$_2$O$_2$, the color of the solution changed from yellow to orange and then to red. The red solution turns dark brown after stirring for 2.5 h at RT. This reaction was strictly performed in a fume hood due to its exothermic nature. After continuous stirring for 2.5 h, 10 ml of ultrapure water was added into the mixture and heated to 60°C overnight to form a brownish gel (V$_2$O$_5$.nH$_2$O). This gel was dried at 100°C for 12 h and subsequently calcined at 400°C for 2 h to get crude VSh. Following this, VSh was probe sonicated in ultrapure water for 2 h to get a dense dispersion of nanosheets. The dispersion was then centrifuged at 1,000 *g* for 5 min and the bright yellow color supernatant was lyophilized to obtain the powdered form of ultrathin V$_2$O$_5$ nanosheets (Vs).

## Characterization of Vs

Powder X-ray diffraction (PXRD) was recorded by Phillips PANalytical diffractometer using a CuKα (λ = 1.5406 Å) radiation. The emission current and accelerating voltage used in the diffractometer were 30 mA and 40 kV, respectively. For morphological and elemental characterization, EDS and scanning electron microscopy (SEM) were performed on FEI Sirion UHR SEM and ESEM-Quanta, respectively. Transmission electron microscopy (TEM), high-resolution transmission electron microscopy (HRTEM), and X-ray mapping images were recorded on JEOL transmission electron microscope operated at 200 kV after casting a drop of nanoparticle dispersion in isopropyl alcohol, over a Cu grid. FT-Raman spectra were recorded using a Renishaw in-Via Raman Microscope

(Renishaw Inc, UK), with excitation wavelength 514 nm. To perform the entire enzyme-mimetic activity assay, SHIMADAZU UV-2600 spectrophotometer was used. X-ray photoelectron spectroscopy (XPS) was performed using AXIS Ultra, KRATOS ANALYTICAL, SHIMADAZU. The surface area measurement was performed by Brunauer–Emmett–Teller (BET) method on the micromeritics surface area analyzer model ASAP 2020.

## GPX—mimicking activity of V$_2$O$_5$ nanoparticles (NPs)

The GPX-like activity of V$_2$O$_5$ NPs was assessed spectrophotometrically by using the standard GR-coupled GPX assay (Vernekar *et al*, 2014). The components and the concentration used in this assay mixture were GSH (2.0 mM), NADPH (0.2 mM), GR 1.7 U, catalyst 20 ng/μl, and H$_2$O$_2$ (0.2 mM) in sodium phosphate buffer pH 7.4 at 25°C. The rate of the reaction was quantified by following the decrease in the absorbance of NADPH (ε = 6220 M$^{-1}$ cm$^{-1}$ at 340 nm) to form NADP$^+$ which is equal to the rate of conversion of H$_2$O$_2$ to H$_2$O.

## Dispersion of V$_2$O$_5$ NPs

V$_2$O$_5$ NPs were dispersed in sterile water at a concentration of 2 mg/ml. The dispersion was carried out by sonication using probe sonicator under the following conditions: time—5 min, amplitude—5 s ON, 5 s OFF. After dispersion, the vanadia NPs form a yellowish colloidal solution.

## Treatment of cell lines with Vs

Indicated cell lines were treated with different concentrations of Vs at a cell density of 0.2 × 10$^6$ cells/ml for 15 min at 37°C in a CO$_2$ incubator. Following Vs treatment, the cells were washed and resuspended in complete medium for further culturing or in FACS buffer (1 × PBS + 3% FBS) for measuring the antioxidant response by flow cytometry, as required. For experiments with J1.1, Jurkat, or CEM-GFP cell lines, which required long-term culturing, Vs treatment was repeated every 24 h, as mentioned above.

## Internalization of Vs by U1 cells

U1 cells were treated with 50 ng/μl and 100 ng/μl of Vs for 15 min and washed thoroughly. Untreated or Vs-treated cells were fixed with 2.5% glutaraldehyde for 16 h followed by gradual dehydration by alcohol gradient. The cells were air-dried, casted on cover slips, and sputtered with gold before being imaged by SEM. Vanadium (V) content was quantified by EDX spectroscopy.

U1 cells were treated with 50 ng/μl Vs as mentioned above. The cells were washed repeatedly to remove cell surface associated Vs and lysed in 10 ml of dilute nitric acid (HNO$_3$; 3 N). The samples are analyzed by ICP-MS (inductively coupled plasma mass spectroscopy), and their (V) content was estimated in parts per billion (ppb). The ppb content of vanadium sulfate treated parallelly with HNO$_3$ was used as a standard to estimate the cellular V content.

## Mammalian cell culture

The human monocytic cell line U937, CD4$^+$ T lymphocytic cell line Jurkat (ATCC, Manassas, VA), the chronically infected U1 and J1.1,

and CD4[+] reporter T-cell line, CEM-GFP (AIDS Research and Reference Reagent program, NIH, USA) were grown in RPMI 1640 (Cell Clone), with 10% FBS (MP Biomedicals) and 2 mM L-glutamine (MP Biomedicals) supplementation. Vs treatment, transfection, and HIV-1 infection were carried out in Opti MEM media (Hyclone). HIV activation in U1 cells was carried out by treatment with 5 ng/ml of phorbol ester PMA (Sigma) or 1.25 μM prostratin (Sigma).

## Preparation of stable cell lines and validation using flow cytometry

Various cell lines stably expressing the cytosolic biosensor Grx1-roGFP2 or Orp1-roGFP2 were prepared by electroporation of $10 \times 10^6$ U1 cells with 5 μg of the pMSCVpuro-Grx1-roGFP2 or pMSCVpuro-Orp1-roGFP2 constructs, followed by selection on 350 ng/ml puromycin. The ratiometric responses of the biosensors were measured by excitation at 405 and 488 nm, and recording emission at 510/10 nm, using BD FACSVerse flow cytometer (BD Biosciences). The data were analyzed using FACSuite software (BD Biosciences).

## Assessment of Vs antioxidant activity and redox potential measurement

$0.1 \times 10^6$ untreated and Vs-treated U1-Orp1-roGFP2 or U1-Grx1-roGFP2 cells were exposed to increasing concentrations of $H_2O_2$ – 50 and 100 μM or 50, 100 and 200 μM, respectively, and incubated at RT for 2–3 min. These cells were analyzed by flow cytometry at excitation of 405 nm (V500) and 488 nm (FITC), while the emission was fixed at 510 nm. Ratio of fluorescence intensities at 405/488 was calculated and normalized using a cell permeable oxidant $H_2O_2$ or the reductant DTT to calculate the responsiveness of both the biosensors.

Intracellular redox potential was measured for cells expressing the Grx1-roGFP2 biosensor, as mentioned earlier (Bhaskar *et al*, 2015). Briefly, for each experiment, 100% biosensor oxidation or reduction corresponding to maximal and minimal fluorescence intensity ratios was determined by treatment with 10 mM $H_2O_2$ and 10 mM DTT, respectively. The observed ratios were used to determine the degree of biosensor oxidation and ultimately equated in a modified form of the Nernst equation to obtain the intracellular glutathione redox potential ($E_{GSH}$).

## Dynamic response of U1-Grx1-roGFP2 cells toward oxidative stress

Oxidation–reduction kinetics of the Grx1-roGFP2 biosensor were measured by flow cytometry, as demonstrated earlier (Bhaskar *et al*, 2015). Briefly, the basal redox state of $1 \times 10^6$ U1-Grx1-roGFP2 cells was measured, following which 50 μM $H_2O_2$ was added after 2 min. Biosensor oxidation and the kinetics of its subsequent recovery were monitored. Parallelly, Vs was added to a set of $H_2O_2$ treated cells at the point of maximum oxidation, and recovery of the biosensor in the presence of Vs was noted. After complete recovery of the cells from oxidative insult, both untreated and Vs-treated cells were challenged with another bolus of 50 μM $H_2O_2$, and the biosensor dynamics were monitored by flow cytometry. Percentage oxidation of the Grx1-roGFP2 biosensor was determined by equating maximal oxidation by 10 mM $H_2O_2$ as 100%.

## Survival assay

U1 cells were treated with increasing concentrations of Vs for 15 min and cultured in complete RPMI medium for 24 h. After 24 h, cells were washed, suspended in $1 \times$ PBS, and stained with 3 μM PI for 15 min in the dark. After washing twice with $1 \times$ PBS, cells were analyzed on a flow cytometer using the phycoerythrin (PE) detector (575/26 nm) by excitation at 488 nm.

Survival of primary CD4[+] T cells from ARV-suppressed individuals were measured using Live/Dead Aqua Dead cell stain kit (BD Biosciences) according to manufacturer's protocol. Briefly, ARV or Vs + ARV-treated cells were harvested at day-14 and day-28 post-culture and incubated with 0.1 μM of the dye in dark for 10 min. The cells were washed with $1 \times$ PBS, fixed with 1% paraformaldehyde (PFA), and analyzed on a flow cytometer by excitation at 405 nm.

## Selenium starvation and HIV-1 reactivation

U1-Grx1-roGFP2/U1 cells grown in complete RPMI medium were harvested and washed three times with serum-free RPMI to remove traces of Se. Cells were seeded in 24 well plates and incubated for 30 min, 1 h, and 2 h in serum-free medium. Parallelly, the cells were treated with various concentrations of Vs for 15 min and cultured as mentioned above. 0.5 nM sodium selenite (Se source) was used as a positive control. Cells were harvested at indicated time points, and the biosensor response was measured by flow cytometry. Viral reactivation was analyzed 6 h post-starvation.

## HIV reactivation in U1 cells and RT–qPCR analysis

U1 cells were treated with either 5 ng/ml PMA or 1.25 μM prostratin and incubated at 37°C in a $CO_2$ incubator. Samples were harvested at 6, 12, and 24 h post-activation, and RNA was isolated using the Qiagen RNeasy kit (manufacturer's protocol). cDNA was synthesized using 400 ng RNA by the Bio-Rad iScript cDNA synthesis kit. RT–qPCR was performed using primers against *gag* transcript (a marker for HIV reactivation). Actin was used as an internal control. To inhibit NF-κB pathway, untreated or Vs-treated cells were exposed to 7.5 μM Bay11-7082 (TCI chemicals) for 12 h.

## Determination of the specificity of Vs toward GSH

The specificity of Vs toward GSH as cofactor was determined by modulating cellular GSH levels. U1-Grx1-roGFP2 cells were treated with 0.5 mM of BSO, an inhibitor of GSH biosynthesis, or supplemented with 7.5 or 15 mM GSH for 16 h. Following this, the cells were treated with 50 ng/μl of Vs and challenged with various $H_2O_2$ concentrations for 2 min. The biosensor response in U1 cells was measured by flow cytometry.

## Effect of TrxR inhibition on Vs activity

To evaluate the effect of the Trx/TrxR redox system on the activity of Vs, U1-Grx1-roGFP2 cells were treated with increasing concentrations of TrxR inhibitor, auranofin (125, 250, and 500 nM) for 16 h. Following this, the cells were treated with 50 ng/μl of Vs and

challenged with various $H_2O_2$ concentrations for 2 min. The biosensor response in U1-Grx1-roGFP2 cells was measured by flow cytometry.

## Assessing the effect of Vs on active HIV replication

CD4[+] T-cell line—CEM-GFP and Jurkat—and monocytic cell line U937 were infected with laboratory-adapted HIV-1, NL-4.3 (CXCR4-using virus), and NL-AD8 (CCR5-using virus), respectively. $0.5 \times 10^6$ untreated or Vs-treated cells of each type were suspended in 500 µl of Opti MEM media and infected at multiplicity of infection (moi) 0.1 (NL-4.3) and 1 (NL-AD8), respectively. The cells were incubated at 37˚C for 4 h and mixed intermittently during the infection period. After 4 h, the cells were washed to remove unbound virus and supplemented with complete RPMI media with 10% FBS. To assess the duration of anti-HIV activity of Vs, CEM-GFP cells were subjected to single dose of Vs, or repeated treatments at 24 and 48 h for 6 days. LTR activation was measured at 2, 4, and 6 dpi by flow cytometry. For subsequent experiments, Vs treatment was repeated every 24 h for Jurkat and CEM-GFP cells. CEM-GFP cells were grown till 6 days, and LTR activation was assessed from day 2 to day 6 post-infection by flow cytometry. Viral replication in Jurkat and U937 cells was monitored by RT–qPCR at indicated time points.

## Nanostring gene expression analysis

Expression levels of 185 genes responsive to oxidative stress and HIV infection were analyzed in untreated U1 cells, U1 cells treated with PMA or Vs alone, and a combination of Vs plus PMA. The Nanostring nCounter analysis system was utilized for this purpose. Briefly, the assay was performed with 100 ng of total RNA, isolated from untreated, or treated cells using the Qiagen RNeasy kit. The purity of the RNA was confirmed spectrophotometrically using NanoDrop Lite Spectrophotometer (Thermo Scientific). The nCounter probes are barcoded DNA oligonucleotides complementary to the target mRNA. Hybridization and counting were performed according to the manufacturer's protocol (Kulkarni, 2011) using a customized panel of 185 genes. 6 housekeeping control genes were included in the panel. Data analysis was done using nSolver 4.0. B2M was used as an internal control due to its minimum % CV.

## Subject samples

Peripheral blood mononuclear cells (PBMCs) were collected from three healthy HIV-seronegative donors and three aviremic HIV-seropositive subjects on stable suppressive ART for a minimum of six years. All subjects provided signed informed consent approved by the Indian Institute of Science, and Bangalore Medical College and Research Institute review boards (IHEC No.- 3-14012020). Primary CD4[+] T cells were purified from PBMCs using an EasySep human CD4[+] T-cell isolation kit (Stem Cell Technologies, Canada). The experiments conformed to the principles set out in the WMA Declaration of Helsinki and the Department of Health and Human Services Belmont Report.

## Infection of primary CD4[+] T cells

Primary CD4[+] T cells were cultured for 3 days after isolation in RPMI 1640 supplemented with 10% FBS, 100 U/ml interleukin-2

(IL-2) (Peprotech, London, United Kingdom) (specific activity, 10 U/ng), and 1 µg/ml phytohemagglutinin (PHA) (Thermo Fisher Scientific). Subsequently, 250,000 activated primary CD4[+] T cells were pre-treated with 25 ng/µl Vs for 15 min and infected with 0.05 moi HIV-1 NL-4.3 virus by spinoculation at 1,000 *g* for 90 min at 32℃. Cells were then washed and replenished with complete media containing 100 U/ml IL-2. Vs treatment was repeated every 48 h. To quantify the virion release, supernatant was harvested from infected cells and centrifuged at 400 g for 10 min and virus concentration was estimated by HIV-1 p24 ELISA (Tyagi et al, 2020).

## Expansion of primary CD4[+] T cells from aviremic subjects

Primary CD4[+] T cells from three ART suppressed (Appendix Table S4), aviremic HIV-infected donors were expanded as described earlier (Kessing *et al*, 2017). Briefly, $50 \times 10^6$ PBMCs were thawed and CD4[+] T cells were isolated using EasySep human CD4[+] T-cell isolation kit (Stem Cell Technologies, Canada). Primary CD4[+] T cells were initially activated using 1 µg/ml of phytohemagluttinin (PHA), 100 U/ml IL-2 and irradiated feeder PBMCs from healthy donors in the presence of antiretrovirals (ARVs—100 nM efavirenz, 180 nM zidovudine and 200 nM raltegravir) alone or Vs in combination with ARVs. 25 ng/µl of Vs treatment was done for 15 min and washed every 3[rd] day. After 7 days, CD4[+] T cells were expanded on ARVs alone or Vs + ARVs and 100 U/ml IL-2 for another 2 weeks. For stimulation experiments at 21[st] day, ARVs and Vs + ARVs were washed off and $1 \times 10^6$ cells were treated with 1 µM prostratin for 24 h.

## RNA and DNA isolation, and nested RT–qPCR from primary CD4[+] T cells

$1 \times 10^6$ cells of ARVs or Vs + ARV-treated samples were harvested at day 14 and 21, and 24 h post-prostratin stimulation using Qiagen RNeasy isolation kit and 200 ng of RNA was reverse transcribed by Bio-Rad iScript cDNA synthesis kit. Reverse transcribed cDNA was diluted 10-fold and amplified using primers against HIV LTRs and nested—PCR was performed using primers and probe listed in Appendix Table S5. Serially diluted pNL4.3 plasmid was used to obtain the standard curve. Isolation and RT–qPCR of total HIV DNA were performed as described earlier (Kessing *et al*, 2017). Briefly, $1 \times 10^6$ primary CD4[+] T cells were lysed in 133 µl of lysis buffer 10 mM Tris–HCL, 50 nM KCl, and 400 µg/ml proteinase K. Total HIV DNA content was measured by semi-nested RT–PCR. First amplification performed using Taq polymerase using primers against HIV and CD3 (internal control) for 12 cycles. Nested amplification was done using 10-fold diluted pre-amplified product, probes, and primers listed in Appendix Table S5. Standard curve was established using ACH-2 cell lysate (CD4[+] T-cell line infected with a single copy of HIV-1 genome).

Expression of antioxidant genes—GPX1, GPX4, GSR, and SRXN1, was analyzed at 21-day post-culture. B2 M was used as internal control.

## p24 detection by immunoblotting and ELISA

Untreated or Vs-treated HIV-infected Jurkat cells were harvested at 36, 48, and 72 h post-infection. Cells were lysed in 300 µl of passive lysis buffer (Promega) supplemented with 1 × protease inhibitor

cocktail (Roche). Protein was quantified using Pierce™ BCA Protein Assay Kit (Thermo Fisher Scientific). 50 µg of whole cell lysate (WCL) was mixed with Laemmli buffer, heated at 95°C for 5 min, and was separated on a 12% SDS–PAGE gel. Immunoblotting was performed using primary antibodies against HIV-p24 (Abcam; ab9071) and GAPDH (CST; D4C6R) as an internal control. Horse anti-mouse IgG (CST; 7076) was used as the secondary antibody. For ELISA, we collected supernatant from Jurkat cells infected with HIV at 24, 48, and 72 hpi. p24 levels were determined by sandwich ELISA using the J.J. Mitra's kit as per the manufacturers' instructions. Standard curve was prepared with known amount p24 and utilized for calculating the viral p24 concentration in the medium.

### Apoptosis assay

HIV-1 NL-4.3-infected primary CD4$^+$ T cells were stained with Annexin V/PI staining kit (BD Biosciences) 3 dpi to assess early apoptotic, necrotic, and late apoptotic cells as per the manufacturer's protocol. Briefly, $0.1 \times 10^6$ untreated or Vs-treated, mock, and HIV-infected primary CD4$^+$ T cells were washed with cold PBS. Cells were stained with Annexin V/PI, and apoptotic cells were determined by flow cytometry using FITC (488 nm excitation and 520 nm emission) and phycoerythrin (488 nm excitation and 578 nm emission) channels.

### Isolation and infection of human monocyte-derived macrophages (HMDMs)

Human monocyte-derived macrophages were isolated from PBMCs of healthy donors by plastic adherence and washing of non-adherent cells after 2 days. $5 \times 10^6$ PBMCs were seeded in 6-well plate and differentiated in macrophage media (RPMI 1640 supplemented with 10% FBS, 5% human AB serum [Sigma], 1% penicillin–streptomycin, 2 mM L-glutamine, 1 mM sodium pyruvate and 40 ng/ml M-CSF [BioLegend]) for 7 days. Media was replenished every 3 days. Differentiated macrophages were detached by incubating with 5 mM EDTA at 37°C, in CO$_2$ incubator for 15–20 min. $0.25 \times 10^6$ cells were seeded in 24-well plates overnight in macrophage media. Untreated or 12.5 ng/µl Vs-treated cells were infected with AD8 virus (20 ng of HIV p24 per sample) for 2 h. Vs treatment for 15 min was repeated every 3$^{rd}$ day. To quantify the virion release, supernatant was harvested from infected cells at 7 and 14 dpi and virus concentration was estimated by HIV-1 p24 ELISA.

### Statistical analysis

All statistical analyses were performed using the GraphPad Prism software (Version 8.1). The data values are indicated as mean ± SD. Statistical significance between two non-parametric test groups was determined using the Mann–Whitney rank sum test, unless specified. Analysis of Nanostring data was performed using the nSolver platform. Differences in $P$ values < 0.05 were considered significant.

## Data availability

This study includes no data deposited in external repositories.

**Expanded View** for this article is available online.

---

### The paper explained

#### Problem

Reactivation of latent but replication-competent human immunodeficiency virus (HIV-1) poses a major barrier to curing the infection. Impaired redox metabolism is one of the mechanisms of HIV-1 reactivation; however, application of this knowledge for therapeutic benefits remains challenging due to deleterious side effects of manipulating redox physiology of the infected cells.

#### Results

Here we nanofabricated vanadium pentoxide (V$_2$O$_5$) and organized into ensembles (e.g., nanowires, nanosheets, and ultrathin nanosheets) that efficiently mimic the activity of an antioxidant enzyme; glutathione peroxidase (GPX). The ultrathin nanosheets, by mimicking the activity of GPX, reprograms redox signaling to subvert HIV-1 from monocytes and lymphocytes, without any detrimental consequences. Treatment with ultrathin nanosheets bolsters the antiviral potential of immune cells by reducing the expression of genes involved in virus activation, inflammation, and apoptosis. Importantly, ultrathin nanosheets efficiently blocked viral reactivation in primary CD4$^+$ T cells from ART-suppressed HIV-infected individuals.

#### Impact

Our study establishes the utility of V$_2$O$_5$-based nanozymes for understanding the link between redox metabolism of immune cells, HIV-1, and calls for exploring the therapeutic potential of antioxidant nanozymes against HIV where redox signaling contributes to viral multiplication, latency, and reactivation.

## Acknowledgements

This work was supported by Wellcome Trust-Department of Biotechnology (DBT) India Alliance grant IA/S/16/2/502700 (A.S.) and in part by DBT grants BT/PR13522/COE/34/27/2015, BT/PR29098/Med/29/1324/2018 and BT/HRD/NBA/39/07/2018-19 (A.S.), DBT-IISc Partnership Program grant 22-0905-0006-05-987 436, and the Infosys Foundation. A.S. is a senior fellow of Wellcome Trust-DBT India Alliance. G. M. acknowledges the DST Nano Mission (SR/NM/NS-1380/2014) and SERB (SB/S2/JCB-067/2015), DST, New Delhi for funding. SS and SG acknowledge fellowships from the University Grants Commission (UGC) and Indian Institute of Science (IISc), respectively. We thank Dr. Amit A. Vernekar for helpful discussions. We gratefully acknowledge the NanoString services provided by TheraCUES Innovations Pvt Ltd, Bangalore. We also thank the AFMM Facility, CeNSE, IISc for the microscopic and spectroscopic facilities.

## Author contributions

SS, SG, GM, and AS participated in the design of the study. SS, SG, MM, and VKP carried out the experiments. PS and DTNM contributed in recruiting and isolating PBMCs from HIV-infected subjects. SS, SG, GM, and AS contributed to reagents and analyzed the data. SS, SG, GM, and AS conceived the study, supervised the project, analyzed the data, and drafted the manuscript. All authors read and approved the final manuscript.

## Conflict of interest

The authors have no conflict of interests to declare.

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
