## [Review Process File · EMBO Molecular Medicine]

Antioxidant nanozyme counteracts HIV-1 by modulating intracellular redox potential

Shalini Singh, Sourav Ghosh, Virender Pal, MohamedHusen Munshi, Pooja Shekhar, Diwakar Narasimha Murthy, Govindasamy Mugesh, and Amit Singh

DOI: [10.15252/emmm.202013314](https://doi.org/10.15252/emmm.202013314)

Corresponding author(s): Amit Singh (asingh@iisc.ac.in), Govindasamy Mugesh (mugesh@iisc.ac.in)

Review Timeline:

Submission Date:	26th Aug 20
Editorial Decision:	29th Sep 20
Revision Received:	28th Dec 20
Editorial Decision:	26th Jan 21
Revision Received:	5th Feb 21
Accepted:	19th Feb 21

Editor: Jingyi Hou

Transaction Report:

29th Sep 2020

Dear Prof. Singh,

Thank you for the submission of your manuscript to EMBO Molecular Medicine. We have now received feedback from the three referees whom we asked to evaluate your manuscript. As you will see from the reports below, the referees acknowledge the potential interest of the study. However, they also raise a series of concerns about your work, which should be convincingly addressed in a major revision of the present manuscript.

Without repeating all the points raised in the reviews below, some of the most substantial issues are the following:

- Referees raised a number of serious issues with regard to the adequacy of the cellular models and the mouse model.
- Referees #1 and #3 mentioned the lack of evidence supporting Vs internalization.

During our pre-decision cross-commenting process (in which the referees are given the chance to make additional comments, including on each other's reports), referee #1 suggested a set of essential experiments that should be performed to address the referee concerns and to make the study more conclusive.

"I understand that a number of concerns have been identified by the three referees, and that a lot of work should be done before granting a publication in EMBO Mol Med. The major comments of the referees are in part overlapping in that at least two of them have identified some specific points that the authors should address. I would focus on some fundamental experiments to be suggested to the authors in order to start addressing at least the criticisms that have been raised more than once in the Referees' reports. In this regard, I would suggest that the authors follow a hierarchical pathway to solve the problems residing in their manuscript. By following this pathway, the authors should gradually become aware of the suitability of their paper to the Journal and may interrupt the process as soon as they realize that the pathway gets interrupted.

- 1) The authors should prove, in a model of HIV latency in CD4+ cells derived from ART-treated people living with HIV/AIDS whether preincubation with Vs may decrease a reactivatable viral reservoir, i.e., they could incubate cells with the Vs for 48-72 h in the presence of ART and then try to reactivate the reservoir using well established methods such as PHA or ionomycin or antiCD3/CD28 antibodies. This experiment would address the comments of Referee 1 (poor survival of HIV-infected cells?), Referee 2 (unsuitability of the HIV latency models) and Referee 3 (poor genetic variability of the virus adopted in the experiments).
- 2) The authors should prove the cell association of Vs (this would address the comments of both Referee 1 and 3).
- 3) The authors should show whether Vs increase susceptibility of Vs to cell death of HIV infected cells. This should address the comments of both Referee 1 (decoupling of antioxidant defense mechanisms) and Referee 2 (advantage over ART).
- 4) The authors should use this newly generated evidence to eliminate the part in the animal model/tuberculosis, which has convinced none of the Referees.

5) The authors should repeat their experiments using a primary cell derived macrophage model for HIV replication/latency. This would address the comments of Referees 2 and 3.

6) The authors should then try to address all the other points raised by the single referees, which, at this point, would be minor."

Referees #2 and #3 indicated explicitly that they concur these comments and the to-do list proposed by Referee #1. Referee #2 added "the proposed plan of action is feasible and at the same time it will allow both authors and reviewers to evaluate the progress of the revision. In case the authors cannot answer these points, they should evaluate a possibility of submitting (or transferring) their manuscript to a different journal. "

After discussing with the editorial team, we agree that the list of experiments suggested by referee #1 are required for further consideration of the manuscript at EMBO Molecular Medicine.

Regarding the point #5, we would encourage you to repeat the experiments using macrophages as suggested by Referees #1 and #3. However, this is not mandatory for acceptance. In any case, the potential limitations in this regard should be discussed.

Overall it is clear that publication of the manuscript cannot be considered at this stage. I also note that addressing the referees' concerns in full will be necessary for further considering the manuscript in our journal and this appears to require a lot of additional work and experimentation. I am unsure whether you will be able or willing to address those and return a revised manuscript within the three months deadline. On the other hand, given the potential interest of the findings, I would be willing to consider a revised manuscript with the understanding that the referee concerns must be fully addressed and that acceptance of the manuscript would entail a second round of review. I should remind you that it is EMBO Molecular Medicine policy to allow a single round of revision only and that, therefore, acceptance or rejection of the manuscript will depend on the completeness of your responses included in the next, final version of the manuscript. For this reason, and to save you from any frustrations in the end I would strongly advise against returning an incomplete revision and would also understand your decision if you choose to rather seek rapid publication elsewhere at this stage.

Should you decide to embark in such a revision, revised manuscripts should be submitted within three months of a request for revision; they will otherwise be treated as new submissions, except under exceptional circumstances in which a short extension is obtained from the editor. Also, the length of the revised manuscript may not exceed 60,000 characters (including spaces) and, including figures, the paper must ultimately fit onto optimally ten pages of the journal. Should you find the length constraints to be a problem, you may consider including any peripheral data (but not methods in their entirety) in the form of Supplementary information.

I look forward to seeing a revised form of your manuscript as soon as possible.

Should you find that the requested revisions are not feasible within the constraints outlined here and choose, therefore, to submit your paper elsewhere, we would welcome a message to this effect.

Sincerely,
Jingyi

Jingyi Hou
Editor
EMBO Molecular Medicine

*** Instructions to submit your revised manuscript ***

IMPORTANT: EMBO Molecular Medicine is now Open Access (see Press Release <https://bit.ly/AwAXU5> and updated Author Guidelines <https://www.embopress.org/page/journal/17574684/authorguide>).

*** PLEASE NOTE *** As part of the EMBO Publications transparent editorial process initiative (see our Editorial at <https://www.embopress.org/doi/pdf/10.1002/emmm.201000094>), EMBO Molecular Medicine will publish a Review Process File online to accompany accepted manuscripts. When preparing your letter of response, please be aware that in the event of acceptance, your cover letter/point-by-point document will be included as part of this file, which will be available to the scientific community.

This would include any additional data, figures or scans made available to the editors or referees. If you do NOT want the file to be published or would like to exclude figures, please immediately inform the editorial office via e-mail. More information about this initiative is available in our Instructions to Authors. If you have any questions about this initiative, please contact the editorial office at contact@embomolmed.org.

To submit your manuscript, please follow this link:

<https://embomolmed.msubmit.net/cgi-bin/main.plex>

- 1) a .docx formatted version of the manuscript text (including Figure legends and tables). Please make sure that the changes are highlighted to be clearly visible to referees and editors alike.
- 2) separate figure files*
- 3) supplemental information as Expanded View and/or Appendix. Please carefully check the authors guidelines for formatting Expanded view and Appendix figures and tables at <https://www.embopress.org/page/journal/17574684/authorguide#expandedview>
- 4) a letter INCLUDING the reviewer's reports and your detailed responses to their comments (as Word file).

Also, and to save some time should your paper be accepted, please read below for additional information regarding some features of our research articles:

- 5) The paper explained: EMBO Molecular Medicine articles are accompanied by a summary of the articles to emphasize the major findings in the paper and their medical implications for the non-specialist reader. Please provide a draft summary of your article highlighting
- the medical issue you are addressing,

- the results obtained and
- their clinical impact.

6) For more information: There is space at the end of each article to list relevant web links for further consultation by our readers. Could you identify some relevant ones and provide such information as well? Some examples are patient associations, relevant databases, OMIM/proteins/genes links, author's websites, etc...

7) Author contributions: the contribution of every author must be detailed in a separate section.

8) EMBO Molecular Medicine now requires a complete author checklist (<https://www.embopress.org/page/journal/17574684/authorguide>) to be submitted with all revised manuscripts. Please use the checklist as guideline for the sort of information we need WITHIN the manuscript as well as in the checklist. This is particularly important for animal reporting, antibody dilutions (missing) and exact p-values and n that should be indicated instead of a range.

9) Every published paper now includes a 'Synopsis' to further enhance discoverability. Synopses are displayed on the journal webpage and are freely accessible to all readers. They include a short stand first (maximum of 300 characters, including space) as well as 2-5 one sentence bullet points that summarise the paper. Please write the bullet points to summarise the key NEW findings. They should be designed to be complementary to the abstract - i.e. not repeat the same text. We encourage inclusion of key acronyms and quantitative information (maximum of 30 words / bullet point). Please use the passive voice. Please attach these in a separate file or send them by email, we will incorporate them accordingly.

You are also welcome to suggest a striking image or visual abstract to illustrate your article. If you do please provide a jpeg file 550 px-wide x 400-px high.

10) A Conflict of Interest statement should be provided in the main text

11) Please note that we now mandate that all corresponding authors list an ORCID digital identifier. This takes <90 seconds to complete. We encourage all authors to supply an ORCID identifier, which will be linked to their name for unambiguous name identification.

Currently, our records indicate that the ORCID for your account is 0000-0001-6761-1664.

Please click the link below to modify this ORCID:
Link Not Available

12) The system will prompt you to fill in your funding and payment information. This will allow Wiley to send you a quote for the article processing charge (APC) in case of acceptance. This quote takes into account any reduction or fee waivers that you may be eligible for. Authors do not need to pay any fees before their manuscript is accepted and transferred to our publisher.

Photos 400-800 DPI

Figures are not edited by the production team. All lettering should be the same size and style; figure panels should be indicated by capital letters (A, B, C etc). Gridlines are not allowed except for log plots.

Figures should be numbered in the order of their appearance in the text with Arabic numerals. Each Figure must have a separate legend and a caption is needed for each panel.

***** Reviewer's comments *****

Referee #1 (Comments on Novelty/Model System for Author):

Singh et al. describe that that Vanadium pentoxide nanosheets may prevent HIV-1 from replicating by using cell line models and Primary CD4⁺ T-cells. This work is extremely interesting in that it might provide a link between the two most investigated approaches to cure HIV infection. One is the "shock and kill" approach, aimed at awakening the latent provirus from latency in order to render infected cells recognizable by either the immune system or drugs in order to eliminate them. The other is the "block and lock", aimed at inducing deep viral latency in the infected cells and prevent the virus from replicating. The work is novel and addresses a very important unmet medical need, but the authors should in my opinion provide further experimental information.

1) In order to show activity of Vs, Singh et al. used a GR-coupled assay. Glutathione reductase (GR) is an intracellular enzyme. How can the authors claim that this is the mechanism by which Vs work in vivo and in cell lines? The dimension and chemical nature of Vs, to my knowledge, does not allow penetration into cells. How do they explain this? Can electron microscopy or another technique help to understand how these nanosheets associate with cells?

2) The authors claim that Vs are an antioxidant strategy because they can catalyze conversion of H₂O₂. However, according to the mechanism that they described, Vs would consume intracellular reduced glutathione (GSH) if the GSH decrease were not compensated by GR. But GR uses NADPH as a cofactor, and the activity of Vs in cells would then result in consumption of NADPH in favour of its oxidized form NADP⁺. Consumption of NADPH would in the long term "paid for" by the cell in terms of reduced thioredoxin (Tx) reducing activity. Tx is reconstituted by thioredoxin reductase (TxR), which uses NADPH as a cofactor as well. In this manner the cell would have an advantage in terms of immediate H₂O₂ consumption, but would also have a compromised ability to react to further oxidative stress, as the second main antioxidant system (the Tx/TxR axis) would have a lower ability to reconstitute itself. The authors should decide experiments able to further clarify this issue.

3) Shytaj et al. recently published in a sister journal that one similar oxidative stress decoupling may induce the selective killing of infected cells. Using an iron chelator, they described the selective killing of infected cells: iron (Fe^{2+}) catalyzes through the Fenton reaction the conversion of hydrogen peroxide to more active reactive oxygen species (ROI). This prevents the establishment of an effective antioxidant response and leads to cell death. In this regard, iron chelation can be seen as similar to the activity of Vs that the authors claim, as it prevents an H_2O_2 effect downstream. The authors should investigate in their primary primary $\text{CD}4^+$ T-cell model whether inhibition of viral replication is connected with increased death of the infected cells.

4) The same paper as quoted above shows that an increase of the antioxidant response is linked to cell survival following HIV infection. Does the initial H_2O_2 consumption induced by Vs impair this antioxidant defense in the long term in the primary cell model that the authors adopted?

5) The authors show a positive effect of Vs on antiapoptotic genes. This is extremely interesting, as a cure of HIV/AIDS has recently been obtained using an antiapoptotic agent (Diaz et al.2020). Similarly a case report from Sharon Lewin's group showed that apoptosis may contribute to maintenance of HIV latency. This could be an argument in strong favour of the authors' hypothesis. The authors should thus, in their primary cell system, show whether cells die by necrosis or apoptosis in the presence of Vs.

6) The final part on the murine model is in my opinion very artificial in that the Authors use a defective HIV construct and extend their considerations to another pathogen, i.e. M. tuberculosis. This part can simply be removed and published separately.

Referee #2 (Comments on Novelty/Model System for Author):

Models of latency are definitely not up to date. They use U1 cellular model of latency and lack to use the more physiological models of latency which would be extremely relevant for the study. There have been a number of studies published in the recent years exploring ex vivo primary human T cell models of latency, which would be more relevant for this study than the cancer cell line U937 and its U1 derivative.

Referee #2 (Remarks for Author):

The authors propose an interesting application of nano materials to reduce oxidative stress in the setting of HIV-1 infection or coinfection with Mycobacterium tuberculosis. The validations of the intracellular antioxidant effects of V_2O_5 nanosheets are well designed and conducted. However, the tools used to evaluate therapeutic effects are not up-to-date and unconvincing. In vitro, the

authors show a reduction in viral reactivation in a cell line and reduction of productive infection in a reporter system of primary cells. Both effects can be achieved at a much more potent level by simply adding antiretroviral drugs, and there is no obvious way through which V2O5 nanosheets could improve or even match the antiviral effect of current therapies. They use primary CD4 T cells which they pretreat with Vs to confer anti-viral response in primary cells and measure p24 by ELISA in cell supernatant. They claim that pretreatment with Vs reduces viral loads by 5 and 2.5 fold at 3 and 5 days post infection. While in U1 they are looking at reactivation from latency, here the effect of Vs is supposedly on the establishment of latency, and those are two distinct events from the point of view of molecular mechanisms. This should be further explored, supported with the primary models of latency. There has been significant progress in the field in the recent years, as several groups have described useful models of latency (Bosque/Planelles, S. Lewin, E. P. Browne, Lusic and Shytaj etc).

In vivo, the authors use a mouse model infected with a non-replicating virus. Several alternative and well established models exist which allow evaluating possible effects of treatments on the viral reservoir (which is lacking throughout the paper). Viral reservoir elimination is broadly seen as the gold standard of HIV-1 cure research, but this is not mentioned in the paper.

The authors dedicate several sentences in the introduction to the putative HIV GPX gene. It is not clear why they elaborate on this concept, as it does not seem to have any further implications on their work. On the other hand, the paper fails to mention entire lines of research which have advanced to the clinical stage and which have opposite paradigms (shock and kill, pharmacologic inhibition of antioxidant enzymes). They don't seem to take into account other antioxidant enzymes like TrxReductase, which reconstitutes Trx.

In summary, I would have expected the authors to perform more detailed studies on the effect of Vs on HIV-1 latency instead of expanding to the model of Tuberculosis.

Referee #3 (Comments on Novelty/Model System for Author):

The manuscript can be strengthened by using primary macrophages as the effect of ROS on HIV infection differ between primary macrophages and transform cell lines such as U1.

Referee #3 (Remarks for Author):

In this manuscript, authors characterized the ability of vanadium pentoxide (V2O5) nanosheets (Vs) to modulate ROS and to block HIV replication and reactivation in transformed cell lines. PHA-activated CD4+ T cells (only one donor) were used in one experiment for HIV replication. Using U1 cells with HIV proviral DNA, authors also showed that Vs blocked Mt infection and modulated immune response in vitro. Finally, authors attempted to demonstrate in vivo function of Vs in HIV Tg26 transgenic mice. The data regarding the chemistry of Vs are strong but there are some concerns regarding the results on HIV infection. Overall, the study is interesting, although PMA-activated U1 cells cannot represent primary macrophages. Authors should note this limitation. The followings are specific comments.

1. Line 77-84, ROS can be modulated by many factors including cytokines and other mediators. HIV genome only plays a partial role. If authors wish to emphasize HIV genome, more relevant clinical

isolates should be included in the study. The whole study was done by using a laboratory adaptive strain X4 virus HIV-1NL-43.

2. Line 201, there is no evidence for the uptake of Vs by U1 cells. In Fig 3A, authors showed cell-associated Vs (in the pellet fraction). It's not clear whether Vs were internalized.

3. Is the HIV inhibitory effect of Vs reversible?

4. In Fig 7, the effect of Vs on cells in the wt mice should be included.

5. It's not clear why authors analyze lung cells in HIV Tg26 mice as these mice are mainly used for HIV-associated nephropathy. While the data were interesting, it would be more relevant to see whether Vs reduces tissue pathology in vivo.

6. It's apparent that authors can measure virus production by determining HIV p24 levels. It's not clear why author used RT-PCR in Fig 4 and Fig 7. Viral transcripts do not always represent virus production.

7. Line 292, the effect of ROS on HIV replication is cell-type dependent. Unlike the transformed cell lines, ROS inhibits HIV replication in primary macrophages (see Tasker C et al. JCI insight, 2016).

8. Line 294, CEM-GFP is a cell line, not a clone.

9. Error bars are missing in Fig 5A, 5C and 5F.

10. Please refer HIV-1NL-43 instead of pNL-43, which is the name for the plasmid of this molecular clone.

11. Line 296 and elsewhere, it should read HIV-1NL-43, which is NOT "provirus." Please make corrections throughout the manuscript.

12. The study for HIV Tg26 mice is not required to be conducted in the BSL3 facility. It may mislead the readers.

Manuscript number: EMM-2020-13314

Title: Antioxidant nanozyme counteracts HIV-1 by modulating intracellular redox potential

Dear Dr. Hou,

We thank the reviewers for the encouraging response to our manuscript **“Antioxidant nanozyme counteracts HIV-1 by modulating intracellular redox potential”**. We are grateful to the reviewers for their insightful comments and for providing us with a consolidated list of major experiments for revising the manuscript. The to-do list prepared by reviewers greatly helped us in strengthening our manuscript. We have now added substantial new experimentation that we hope addresses the concerns raised by the reviewers.

Response to the to-do list of experiments proposed by the reviewers'

The authors should prove, in a model of HIV latency in CD4+ cells derived from ART-treated people living with HIV/AIDS whether preincubation with Vs may decrease a reactivatable viral reservoir, i.e., they could incubate cells with the Vs for 48-72 h in the presence of ART and then try to reactivate the reservoir using well established methods such as PHA or ionomycin or antiCD3/CD28 antibodies. This experiment would address the comments of Referee 1 (poor survival of HIV-infected cells?), Referee 2 (unsuitability of the HIV latency models) and Referee 3 (poor genetic variability of the virus adopted in the experiments).

This is an excellent suggestion by the reviewer. Accordingly, we examined the influence of Vs on viral reactivation in latently infected CD4+ T cells isolated from three HIV-infected subjects on suppressive antiretroviral therapy (ART) for a minimum of six years (Kessing *et al*, 2017). The CD4+ T cells were initially expanded in the presence of interleukin-2 (IL-2), phytohemagglutinin (PHA), "feeder cells," or with antiretrovirals (ARVs) alone or ARVs together with Vs. From day 7 onwards, CD4+ T cells were cultured in a medium containing IL-2 and ARVs (ARVs alone) or with ARVs and Vs combination (Vs + ARVs) (Fig 1A, B [Fig 7A, B in the revised manuscript]). The viral transcription increased initially, followed by low to undetectable levels by day 21 (Fig [Fig. in the revised manuscript]). Interestingly, viral RNA levels from all three subjects' cells exposed to Vs + ARVs were below detection compared to only one subject's cell in the case of ARVs alone.

On day 21, we determined the effect of Vs on reactivatable viral reservoir by stimulating the CD4+ T cells with the PKC activator prostratin in the absence of any treatment (Fig 1B [Fig 7B in the revised manuscript]). The activation of viral transcription was measured 24 h later by qRT-PCR. When ARVs were removed, followed by prostratin stimulation, the viral transcript was detected in all subjects' cells (Fig 1C, D [Fig 7C, D in the revised manuscript]). In

contrast, upon Vs + ARVs removal followed by prostratin stimulation, viral transcription was inhibited by 90%, 100%, and 100% from three subjects, respectively, with an average inhibition of 96.7% for all three subjects in 24 h (Fig 1C, D [Fig 7C, D in the revised manuscript]).

Figure 1. Vs blocks viral reactivation in CD4⁺ T cells isolated from virally suppressed patients

A. Schematic representation of generation of expanded CD4⁺ T cells and reactivation. CD4⁺ T cells were sorted from PBMCs of ARVs suppressed HIV-infected individuals and expanded in presence of PHA, IL-2, and autologous feeder PBMCs from healthy donor.

B. Expanded CD4⁺ T cells from 3 patients were cultured in presence of IL-2 and ARVs, with and without 25 ng/μL Vs for 21 days. Vs treatment was given for 15 min every 3rd day. HIV transcripts were quantified by qRT-PCR at day 14, day 21, and at 24 h post-stimulation of cells cultured for 21 days by prostratin. Limit of detection qRT-PCR was 3 viral transcripts per million cells.

C. At day 21, cells were stimulated with 1.25 μM prostratin for 24 h and HIV transcripts were quantified by qRT-PCR. Reduction in viral stimulation in Vs treated samples are represented as percentage values. ND – non determined.

D. Aggregate plot for 3 patients from data (C).

E. Total HIV-1 DNA was determined up to 21 days in cells treated with ARVs or Vs + ARVs.

* P<0.05, ns – non-significant by 2-way ANOVA with Tukey's multiple comparison.

Importantly, we measured the total HIV DNA to confirm that the reduction in viral reactivation by Vs is not due to the selective loss or poor survival of HIV

infected cells. The total HIV DNA content was comparable between the freshly isolated patient's CD4⁺ T cells (*ex vivo*) and the expanded cells treated with ARVs alone or Vs + ARVs combination for the experiment's entire duration (Fig 1E [Fig 7E in the revised manuscript]). Therefore, Vs inhibited viral transcription without affecting proviral content. We trust that these findings resolved the limitations associated with cell line models of latency, clonal HIV laboratory strains, and survivability of HIV-1 infected cells. I want to point out that these experiments were particularly challenging in the current pandemic situation as very few HIV patients visiting ART clinics and inaccessibility of already overburdened clinicians. All these precluded experiments on a large number of patient samples. Notwithstanding these limitations, we very much appreciate the reviewers for this comment.

2) The authors should prove the cell association of Vs (this would address the comments of both Referee 1 and 3).

We apologize for the inadequate explanation, leading to this confusion. We reported Vs nanomaterial's internalization using an ultrasensitive technique called inductively coupled plasma-mass spectrometry (ICP-MS). The U1 cells were treated with 50 ng/ μ l of Vs for 15 min, followed by extensive washing to remove Vs associated with the cell surface. The cells were then lysed, and the lysate was subjected to (ICP-MS) to examine Vs internalization. Using ICP-MS, we clearly identified buildup of Vs inside U1, followed by a gradual decrease over time such that only a fraction of internalized Vs was retained (Fig 3A in the revised manuscript). We have replaced the word "cell pellet" with "cell lysate" in figure 3A of the revised manuscript for clarity and clearly described the procedure in the methods section.

In addition to ICP-MS, we convincingly showed that the glutathione peroxidase (GPX) activity of Vs depends on the intracellular redox buffer glutathione (GSH). Inhibition of GSH biosynthesis via buthionine sulfoximine (BSO) abrogated, whereas GSH supplementation restored the GPX activity of Vs (Fig 3G, H of the revised manuscript). Importantly, we verified the antioxidant activity of Vs by measuring intracellular hydrogen peroxide (H₂O₂) and glutathione redox potential (E_{GSH}) using orp1-roGFP2 and Grx1-roGFP2 biosensors, respectively (Fig 3C, E of the revised manuscript). Both sensors were genetically expressed in the cytosol of the U1 cells, requiring internalization of Vs for measuring dynamic changes in H₂O₂ and E_{GSH} .

To further address this issue, we investigated the internalization of Vs using scanning electron microscopy (SEM) coupled with electron dispersive X-ray spectroscopic (EDS). The surface of Vs treated U1 cells exhibited depressions and indentations, indicative of Vs uptake. We confirmed this by carrying out EDS analysis of concavity, which identified the peak corresponding to vanadium specifically in Vs-treated cells. These results are incorporated in the revised manuscript (Fig 2A-C [Fig EV3A-C in the revised manuscript]).

Figure 2. EDS coupled scanning electron microscopy analysis of Vs treated U1 cells.

A-C. U1 cells were either left untreated (A) or treated with (B) 50 and (C) 100 ng/μL of Vs for 15 min and immediately harvested followed by fixation, dehydration, and imaging by scanning electron microscopy. Left panel shows depression (marked by arrows) formed on the cell surface (B and C). Scale bar 3μm (A) and (B), 5μm (C). *Right panels:* The depressions on cell surfaces due to Vn internalization were verified by EDS analysis on the depressed regions. The EDX plots showing the absence or presence of vanadium peak in untreated and Vs-treated U1 cells, respectively.

The authors should show whether Vs increase susceptibility of Vs to cell death of HIV infected cells. This should address the comments of both Referee 1 (decoupling of antioxidant defense mechanisms) and Referee 2 (advantage over ART).

This concern is addressed in our experiment on latently infected CD4+ T cells derived from HIV patients on suppressive ART. These cells were exposed to Vs for 21 days to assess the effect on reactivation upon prostratin stimulation. In parallel, we measured total HIV DNA using qPCR. We found no decrease in the total HIV DNA content between the freshly isolated patient's CD4+ T cells (labelled *ex vivo*) and the expanded cells treated with ARVs alone or Vs + ARVs for the entire duration of the experiment (refer to figure 1E [Fig. 7E in

the revised figure]. Since the HIV DNA content between freshly isolated *ex vivo* cells remain comparable to expanded cells, suggesting that the characteristics of an individual's viral reservoir remain preserved upon Vs treatment. Altogether, data suggest that Vs treatment does not affect the survival of HIV infected cells.

Decoupling antioxidant response: Our findings suggest that decoupling of antioxidant defense mechanisms is counterbalanced by the efficient GPX and anti-HIV activities of Vs. Since HIV-1 proteins such as gp120, Nef, Tat, Vpr, and reverse transcriptase elevate ROS production and induce Nrf2/ARE pathway (Ivanov *et al*, 2016; Mastrantonio *et al*, 2016), the inhibitory effect of Vs on HIV replication would have additionally contributed to the reduction in ROS levels and Nrf2/ARE pathway by viral products.

We tested the frequency of primary CD4⁺ T cells infected with HIV-1 NL-4.3 for markers of early apoptosis, late apoptosis, and necrosis upon Vs treatment to further address this point. The treatment of HIV-1 infected human CD4⁺T cells with Vs significantly reduces the fraction of cells exhibiting late apoptosis. Also, the frequency of early apoptotic or necrotic cells was not increased by Vs (Fig 3A [Fig EV5B in the revised manuscript]). Our findings are consistent with the requirement of robust antioxidant potential of cells chronically infected with HIV in resisting apoptosis and cell death (Bhaskar *et al*, 2015; Fernandez Larrosa *et al*, 2008; Pinti *et al*, 2003). Agreeing to this, efficient inhibition of the Trx/TrxR antioxidant system by a pro-oxidant gold-based drug auranofin allowed targeting of HIV reservoirs by exerting pro-differentiating and pro-apoptotic effects (Chirullo *et al*, 2013). Using a combination of auranofin, BSO, and ART, Shytaj *et al.*, reported complete clearance of SIV viremia in macaques with 100% AIDS-free survival for at least 2 years after interruption of therapy (Shytaj *et al*, 2013; Shytaj *et al*, 2015). These studies underscore the importance of studying redox metabolism in controlling HIV infection. Our results indicate that Vs' robust antioxidant, cytoprotective, and anti-HIV properties might provide an important alternative to HIV cure by blocking viral reactivation and locking in a state of sustained latency. Future experiments will investigate the underlying mechanism (genetic/epigenetic) and how long Vs extends suppression of HIV transcription in combination with ART or even after treatment interruption.

Figure 3. Vs inhibits HIV-1 replication in HMDMs and reduces late apoptosis in HIV-infected cells

A. Survival of HIV-infected primary CD4⁺ T cells was monitored by Annexin V/PI staining at 3 dpi in presence or absence of Vs treatment. Percentage of necrotic (PI⁺), early apoptotic (Annexin V⁺) and late apoptotic (Annexin V⁺/PI⁺) cells under different conditions are plotted. ** P<0.01, ns – non-significant analysed by 2-way ANOVA. Data are aggregate from three healthy donors (mean ± SEM).

Advantage over ART: The clinical relevance comes from our results indicating a better ability of Vs + ARVs in inhibiting viral transcription during the expansion phase compared to treatment with ARVs alone. However, robustness and significance of this trend needs a large sample size. More importantly, Vs blocks virus rebound upon stimulation with prostratin. These findings raise the possibility of including Vs with the frontline treatment for faster suppression and potentially reducing the reservoir's size. It has been suggested that transient viral replication ("blips") observed in plasma could be reseeding the reservoir even in the presence of ART (Jones & Perelson, 2007; Ramratnam *et al*, 2004). Furthermore, therapy non-compliance can also result in viremia and reservoir replenishment. Combining Vs in ART regimens could potentially inhibit reservoir replenishment during these situations. Lastly, clinically relevant anti-retroviral drugs have been shown to induce massive ROS (Mondal *et al*, 2004; Weiss *et al*, 2016), which may contribute to the development of cardiovascular diseases and CNS pathologies (Hurwitz *et al*, 2004; Masia *et al*, 2007; Opii *et al*, 2007). The inclusion of antioxidant nanozymes such as Vs could help alleviate ART-induced ROS production to improve therapy outcomes. Our study introduced nanozymes as future platforms to develop interventions against HIV latency and reactivation. These points are included in the discussion section of the revised manuscript.

We thank the reviewers for these insightful comments.

The authors should use this newly generated evidence to eliminate the part in the animal model/tuberculosis, which has convinced none of the Referees.

As suggested by the referees, we have removed the data related to HIV-Tg animals and tuberculosis.

The authors should repeat their experiments using a primary cell derived macrophage model for HIV replication/latency. This would address the comments of Referees 2 and 3.

We also examined if the antioxidant potential of Vs confers anti-viral response in primary human macrophages. Monocyte-derived macrophages (MDMs) were differentiated from human PBMCs for 7 days in the presence of 40 ng/mL M-CSF followed by pretreatment with 12.5 ng/μL of Vs for 15 min and

infection with HIV-1 NL-AD8. Viral release in the supernatant was quantified by p24 ELISA at 7- and 14-days post-infection. The p24 ELISA confirmed a time-dependent increase in virus load, which was uniformly reduced upon pretreatment of primary MDMs with Vs (Fig. 4A [Fig. EV5A in the revised manuscript]).

Figure 4. Vs inhibits HIV-1 replication in HMDMs and reduces late apoptosis in HIV-infected cells

A. Human monocyte-derived macrophages (HMDMs) were pre-treated with 12.5 ng/ μ L of Vs for 15 min, followed by infection with HIV-1 NL-AD8. Viral release in supernatant was quantified by p24 ELISA at 7 and 14 dpi. Vs treatment was repeated every 72 h. Data are obtained from one healthy donor (mean \pm SD).

The authors should then try to address all the other points raised by the single referees, which, at this point, would be minor."

A point-by-point response to the minor points raised by reviewers is given below:

Reviewer #1

In order to show activity of Vs, Singh et al. used a GR-coupled assay. Glutathione reductase (GR) is an intracellular enzyme. How can the authors claim that this is the mechanism by which Vs work in vivo and in cell lines? The dimension and chemical nature of Vs, to my knowledge, does not allow penetration into cells. How do they explain this? Can electron microscopy or another technique help to understand how these nanosheets associate with cells?

The reviewer is correct. We examined the H₂O₂ reducing activity (i.e., glutathione peroxidase (GPX)] of Vs using a GR-coupled assay *in vitro*. To show this in cell lines, we expressed a genetic biosensor of H₂O₂ (orp1-roGFP2) in the cytoplasm of U1 and showed that Vs reduces intracellular

H₂O₂ in real-time. Since H₂O₂ oxidizes glutathione (GSH) to glutathione disulfide (GSSG), we expect this transformation to be affected by the Vs-mediated reduction of H₂O₂. Therefore, we expressed a genetic biosensor of glutathione redox potential (E_{GSH}) in the cytosol of U1 and demonstrated that H₂O₂-mediated oxidation of GSH was reduced in the presence Vs. Both of these technologies confirmed that Vs functions as an efficient mimic of GPX inside cells.

The earlier section addressed the penetration issue (please see the response to point #2- to-do-list). We confirmed the internalization of Vs by U1 using ICP-MS. Furthermore, SEM analysis of Vs treated cells shows depressions and indentations, which indicate Vs uptake. We confirmed this by EDS analysis of concavity, which identified the peak corresponding to vanadium specifically in Vs-treated cells (Fig 2A-C [Fig EV3A-C in the revised manuscript]). We thank the reviewer for this comment.

The authors claim that Vs are an antioxidant strategy because they can catalyze conversion of H₂O₂. However, according to the mechanism that they described, Vs would consume intracellular reduced glutathione (GSH) if the GSH decrease were not compensated by GR. But GR uses NADPH as a cofactor, and the activity of Vs in cells would then result in consumption of NADPH in favour of its oxidized form NADP⁺. Consumption of NADPH would in the long term "paid for" by the cell in terms of reduced thioredoxin (Tx) reducing activity. Tx is reconstituted by thioredoxin reductase (TxR), which uses NADPH as a cofactor as well. In this manner the cell would have an advantage in terms of immediate H₂O₂ consumption, but would also have a compromised ability to react to further oxidative stress, as the second main antioxidant system (the Tx/TxR axis) would have a lower ability to reconstitute itself. The authors should decide experiments able to further clarify this issue.

We appreciate the reviewer's comment. We agree that the long-term effect of Vs on Trx/TxR and NADPH needs further experimentation. To begin addressing this issue, we examined if Vs activity is affected by the Trx/TxR system. We pretreated U1-Grx1-roGFP2 cells with various concentrations of the TrxR inhibitor auranofin and examined the ability of Vs to protect biosensor oxidation by H₂O₂. We used auranofin concentrations, which are known to inhibit TrxR inside cells (Cox *et al*, 2008). We found that Vs is fully functional and efficiently mitigates oxidative stress despite the presence of auranofin (Fig 5A [fig EV4A in the revised manuscript]). Since the study is not intended to dissect the effect of the Trx/TxR system on HIV latency/reactivation but rather to explore how Vs modulates redox metabolism and HIV infection, we hope that reviewer would agree with our views that exhaustive analyses of how Vs impacts other redox couples, which would require a large number of well-controlled experiments, should be the focus of an independent study. We thank the reviewer for this comment.

Figure 5. Auranofin does not influence the antioxidant activity of Vs.

A. U1 Grx1-roGFP2 cells were supplemented with increasing doses of auranofin for 16 h to inhibit thioredoxin reductase (TrxR). Following this, cells were treated with Vs for 15 min, exposed to H₂O₂, and the ratiometric response was measured by flow cytometry. Data are representative of results from two independent experiments performed in duplicate (mean ± SEM). ns – non-significant, by Mann Whitney Test.

Shytaj et al. recently published in a sister journal that one similar oxidative stress decoupling may induce the selective killing of infected cells. Using an iron chelator, they described the selective killing of infected cells: iron (Fe²⁺) catalyzes through the Fenton reaction the conversion of hydrogen peroxide to more active reactive oxygen species (ROI). This prevents the establishment of an effective antioxidant response and leads to cell death. In this regard, iron chelation can be seen as similar to the activity of Vs that the authors claim, as it prevents an H₂O₂ effect downstream. The authors should investigate in their primary CD4⁺T-cell model whether inhibition of viral replication is connected with increased death of the infected cells.

We thank the reviewer for this intuitive comment. The study by Shytaj et al., demonstrated that long-term treatment with low concentrations of iron chelator deferiprone induced the killing of HIV-infected cells. However, whether the increased mortality is due to lack of hydroxyl radical formation via Fenton reaction and inefficient antioxidant response remains tested. While the study does imply this possibility, it is well known that iron limitation also compromises the activity of Fe-S cluster enzymes (e.g., aconitase, succinate dehydrogenase) coordinating mitochondrial metabolism (TCA) and heme-containing respiratory enzymes (cytochrome oxidases) involved in oxidative phosphorylation (OXPHOS) (Oexle *et al*, 1999). Therefore, along with the perturbed antioxidant response, Fe-limitation could modulate central metabolism and OXPHOS. Both central carbon metabolism and OXPHOS are well known to affect virus reactivation, reservoir seeding, and killing of HIV infected cells (Castellano *et al*, 2019; Tyagi *et al*, 2020; Valle-Casuso *et al*, 2019).

To further address this, we investigated the influence of Vs on death of latently infected CD4⁺ T cells derived from HIV patients on suppressive ART and also examined apoptosis and necrosis in primary CD4⁺ T cell (please see response to point # 3, to-do-list). Both of these assays ruled out increased mortality of HIV infected cells by Vs, consistent with our earlier report on cytoprotective properties of vanadia nanowires (Vernekar *et al*, 2014). However, future experimentations are needed to investigate the long-term effect of Vs on the viability of HIV infected cells.

The authors show a positive effect of Vs on antiapoptotic genes. This is extremely interesting, as a cure of HIV/AIDS has recently been obtained using an antiapoptotic agent (Diaz et al.2020). Similarly a case report from Sharon Lewin's group showed that apoptosis may contribute to maintenance of HIV latency. This could be an argument in strong favour of the authors' hypothesis. The authors should thus, in their primary cell system, show whether cells die by necrosis or apoptosis in the presence of Vs.

As per the reviewer's suggestion, we tested the influence of Vs on apoptosis and necrosis in the primary CD4⁺ T cell model of viral replication (please see the response to point # 3, to-do-list). Consistent with our expression data, we found that Vs pretreatment decrease late apoptosis and does not affect the frequency of cells undergoing necrosis. These findings raise the possibility of exploring antioxidant properties of Vs with the current ART regimen to reduce virus reactivation from latent reservoirs- a strategy similar to the "block-and-lock" approach. We thank the reviewer for this suggestion.

The final part on the murine model is in my opinion very artificial in that the Authors use a defective HIV construct and extend their considerations to another pathogen, i.e. *M. tuberculosis*. This part can simply be removed and published separately.

We have removed animal data and experiment with *M. tuberculosis* from the revised manuscript.

Reviewer # 2

The authors propose an interesting application of nano materials to reduce oxidative stress in the setting of HIV-1 infection or coinfection with *Mycobacterium tuberculosis*. The validations of the intracellular antioxidant effects of V2O5 nanosheets are well designed and conducted. However, the tools used to evaluate therapeutic effects are not up-to-date and unconvincing. In vitro, the authors show a reduction in viral reactivation in a cell line and reduction of productive infection in a reporter system of primary cells. Both effects can be achieved at a much more potent level by simply adding antiretroviral drugs, and there is no obvious way through which V2O5 nanosheets could improve or even match the antiviral effect of current therapies. They use primary CD4 T cells which they pretreat with Vs to confer anti-viral response in

primary cells and measure p24 by ELISA in cell supernatant. They claim that pretreatment with Vs reduces viral loads by 5 and 2.5 fold at 3 and 5 days post infection. While in U1 they are looking at reactivation from latency, here the effect of Vs is supposedly on the establishment of latency, and those are two distinct events from the point of view of molecular mechanisms. This should be further explored, supported with the primary models of latency. There has been significant progress in the field in the recent years, as several groups have described useful models of latency (Bosque/Planelles, S. Lewin. E. P. Browne, Lusic and Shytaj etc).

We thank the reviewer for this comment. As suggested by the reviewer, we now include data from the primary CD4⁺ T cell models of latency to show that Vs exerts additive effect with ARVs in inhibiting viral transcription during replication phase and blocks virus rebound upon stimulation with prostratin when ARVs were removed (please see the response to point#1, to-do-list). Additionally, we included data of primary human CD4⁺ T cells and human monocyte-derived macrophages (HMDM), demonstrating that Vs inhibits viral replication (Fig 6F and Fig EV5A in the revised manuscript).

In vivo, the authors use a mouse model infected with a non-replicating virus. Several alternative and well established models exist which allow evaluating possible effects of treatments on the viral reservoir (which is lacking throughout the paper). Viral reservoir elimination is broadly seen as the gold standard of HIV-1 cure research, but this is not mentioned in the paper.

As suggested by the reviewer, we have removed the findings obtained using the mouse model. Further, we discussed the importance of viral reservoir elimination and the anti-HIV potential of Vs in inhibiting reservoir replenishment during therapy non-compliance and transient viremia. We thank the reviewer for this comment.

The authors dedicate several sentences in the introduction to the putative HIV GPX gene. It is not clear why they elaborate on this concept, as it does not seem to have any further implications on their work. On the other hand, the paper fails to mention entire lines of research which have advanced to the clinical stage and which have opposite paradigms (shock and kill, pharmacologic inhibition of antioxidant enzymes). They don't seem to take into account other antioxidant enzymes like TrxReductase, which reconstitutes Trx.

We apologize for the inadequate description. In the revised manuscript, we have reduced the HIV GPX gene's description and included studies about the Trx/TrxR system in the introduction.

Reviewer # 3

The manuscript can be strengthened by using primary macrophages as the effect of ROS on HIV infection differ between primary macrophages and transform cell lines such as U1.

In the revised manuscript, we have included findings on primary macrophages. Monocyte-derived macrophages (MDMs) were differentiated from human PBMCs for 7 days in the presence of 40 ng/mL M-CSF followed by pretreatment with 12.5 ng/ μ L of Vs for 15 min and infection with HIV-1 AD8. Viral release in the supernatant was quantified by p24 ELISA at 7 and 14 days post-infection. The p24 ELISA confirmed a time-dependent increase in virus load, which was uniformly reduced upon pretreatment of primary MDMs with Vs (Fig. 4A [Fig. EV5A in the revised manuscript]). Our findings are in agreement with studies showing the role of ROS and reactive nitrogen intermediates (RNI) in supporting HIV replication in primary macrophages (Aquaro *et al*, 2007).

In this manuscript, authors characterized the ability of vanadium pentoxide (V₂O₅) nanosheets (Vs) to modulate ROS and to block HIV replication and reactivation in transformed cell lines. PHA-activated CD4⁺ T cells (only one donor) were used in one experiment for HIV replication. Using U1 cells with HIV proviral DNA, authors also showed that Vs blocked Mt infection and modulated immune response in vitro. Finally, authors attempted to demonstrate in vivo function of Vs in HIV Tg26 transgenic mice. The data regarding the chemistry of Vs are strong but there are some concerns regarding the results on HIV infection. Overall, the study is interesting, although PMA-activated U1 cells cannot represent primary macrophages. Authors should note this limitation. The followings are specific comments.

We thank the reviewer for these comments. In the revised manuscript, we have included data showing the effect of Vs on the replication of HIV using CD4⁺ T cells from three donors (Fig 6A, [Fig 6F in the revised manuscript]). Furthermore, we performed an experiment using primary macrophages and demonstrated anti-HIV potential of Vs (Fig 4A, [Fig EV5A in the revised manuscript]).

Figure 6. Vs reduces replication of HIV-1 in primary CD4⁺ T cells.

A. Primary CD4⁺ T cells purified from human PBMCs (3 healthy donors) were activated, pre-treated with 25 ng/μL Vs for 15 min, and infected with 0.05 moi of HIV-1 NL-4.3. Virus released in supernatant was quantified by p24 ELISA. Vs treatment was repeated every 48 h. All figures were analysed by 2-way ANOVA. **** P<0.0001, ** P<0.01, * P<0.05. Data represents aggregate results from three healthy donors (mean ± SEM).

Line 77-84, ROS can be modulated by many factors including cytokines and other mediators. HIV genome only plays a partial role. If authors wish to emphasize HIV genome, more relevant clinical isolates should be included in the study. The whole study was done by using a laboratory adaptive strain X4 virus HIV-1NL-43.

This issue is now resolved with the use of latently infected CD4⁺ T cells derived from 3 aviremic HIV-subjects of suppressive ART. We thank the authors for this comment.

Line 201, there is no evidence for the uptake of Vs by U1 cells. In Fig 3A, authors showed cell-associated Vs (in the pellet fraction). It's not clear whether Vs were internalized.

We apologize for the confusion. Figure 3 A shows Vs inside the cells (not pellet). This has been corrected, and the concern is fully addressed earlier (please see the response to point#, to-do-list).

Is the HIV inhibitory effect of Vs reversible?

CD4⁺ T cell line expressing EGFP (CEM-GFP) under HIV-1 LTR, we showed that the infection with T cell-tropic virus (HIV-1 NL-4.3) progressively increased GFP fluorescence over 5 days (Fig 6A, in the revised manuscript). Addition of 50 ng/μL of Vs for 15 min every 24 h completely blocked GFP expression in the infected CEM-GFP cells (Fig 6A in the revised manuscript). However, exposure to only a single dose of Vs for 15 min followed by washout did not inhibit GFP expression, whereas 15 min of Vs exposure every 48 h partially reduced GFP expression (Appendix Fig S6A in the revised manuscript). These results indicate that repeated treatment of Vs for 15 min every 24 h interval is required to sustain the inhibitory effect on HIV-1 replication.

In Fig 7, the effect of Vs on cells in the wt mice should be included.

5. It's not clear why authors analyze lung cells in HIV Tg26 mice as these mice are mainly used for HIV-associated nephropathy. While the data were interesting, it would be more relevant to see whether Vs reduces tissue pathology in vivo.

We thank the reviewer for above suggestions. However as per the advice of other reviewers, we have removed the data related to animal studies.

6. It's apparent that authors can measure virus production by determining HIV p24 levels. It's not clear why author used RT-PCR in Fig 4 and Fig 7. Viral transcripts do not always represent virus production.

We agree with the reviewer. The choice of RT-PCR was used to assess the antioxidant effect of Vs as HIV transcription is controlled by multiple redox-responsive transcription factors (NF- κ B, AP-1, and NFAT) (Pyo *et al*, 2008). Additionally, we complemented our findings with p24 ELISA and immunoblotting. In the revised manuscript, the effect of Vs on HIV replication in primary CD4⁺ T cells and monocyte-derived macrophages was determined by measuring virus production (Fig 6A and 4A [Fig 6F and Fig EV5A in the revised manuscript]).

7. Line 292, the effect of ROS on HIV replication is cell-type dependent. Unlike the transformed cell lines, ROS inhibits HIV replication in primary macrophages (see Tasker C et al. JCI insight, 2016).

We thank the reviewer for this comment. The effect of ROS on HIV replication in primary macrophages needs further experimentation. The study by Tasker *et al.* indicated that ROS generators paraquat and plumbagin inhibit HIV replication. While interesting, ROS production by paraquat/plumbagin is dependent on mitochondrial inner-transmembrane potential, the presence of respiratory substrates, and functional complex III (Castello *et al*, 2007). Since HIV replication affects mitochondrial activity (Hulgan & Samuels, 2020), the magnitude and type(s) of ROS produced by paraquat/plumbagin remain to be tested. The study by Tasker *et al.*, also noted that N-acetyl cysteine (NAC) at concentrations (10 mM) known to counteract oxidative stress (Ezerina *et al*, 2018), inhibits HIV replication in primary macrophages. Only at lower concentrations (10 - 30 μ M), NAC partially diminished IFN- ϵ mediated HIV inhibition in Tasker *et al.*, (Tasker *et al*, 2016). Lastly, other studies indicated that ROS and RNI support HIV replication in primary macrophages (Aquaro *et al.*, 2007). Therefore, in our opinion, ROS's issue on HIV replication in the context of primary macrophages needs detailed investigation as suggested by Tasker *et al.* However, we have included the reference of this important contribution in our study.

8. Line 294, CEM-GFP is a cell line, not a clone.

We have corrected this in the revised manuscript.

9. Error bars are missing in Fig 5A, 5C and 5F.

We have included error bars in Fig 5A, 5C and 5F (Fig 6A, 6C, and 6F in the revised manuscript).

10. Please refer HIV-1NL-43 instead of pNL-43, which is the name for the plasmid of this molecular clone.

We have corrected this in the revised manuscript.

11. Line 296 and elsewhere, it should read HIV-1NL-43, which is NOT "provirus." Please make corrections throughout the manuscript.

We have corrected this in the revised manuscript.

12. The study for HIV Tg26 mice is not required to be conducted in the BSL3 facility. It may mislead the readers.

We have removed reference of BSL3 from the revised manuscript.

References

- Aquaro S, Muscoli C, Ranazzi A, Pollicita M, Granato T, Masuelli L, Modesti A, Perno CF, Mollace V (2007) The contribution of peroxynitrite generation in HIV replication in human primary macrophages. *Retrovirology* 4: 76
- Bhaskar A, Munshi M, Khan SZ, Fatima S, Arya R, Jameel S, Singh A (2015) Measuring glutathione redox potential of HIV-1-infected macrophages. *J Biol Chem* 290: 1020-1038
- Castellano P, Prevedel L, Valdebenito S, Eugenin EA (2019) HIV infection and latency induce a unique metabolic signature in human macrophages. *Sci Rep* 9: 3941
- Castello PR, Drechsel DA, Patel M (2007) Mitochondria are a major source of paraquat-induced reactive oxygen species production in the brain. *J Biol Chem* 282: 14186-14193
- Chirullo B, Sgarbanti R, Limongi D, Shytaj IL, Alvarez D, Das B, Boe A, DaFonseca S, Chomont N, Liotta L *et al* (2013) A candidate anti-HIV reservoir compound, auranofin, exerts a selective 'anti-memory' effect by exploiting the baseline oxidative status of lymphocytes. *Cell Death Dis* 4: e944
- Cox AG, Brown KK, Arner ES, Hampton MB (2008) The thioredoxin reductase inhibitor auranofin triggers apoptosis through a Bax/Bak-dependent process that involves peroxiredoxin 3 oxidation. *Biochem Pharmacol* 76: 1097-1109
- Ezerina D, Takano Y, Hanaoka K, Urano Y, Dick TP (2018) N-Acetyl Cysteine Functions as a Fast-Acting Antioxidant by Triggering Intracellular H₂S and Sulfane Sulfur Production. *Cell Chem Biol* 25: 447-459 e444
- Fernandez Larrosa PN, Croci DO, Riva DA, Bibini M, Luzzi R, Saracco M, Mersich SE, Rabinovich GA, Martinez Peralta L (2008) Apoptosis resistance in HIV-1 persistently-infected cells is independent of active viral replication and involves modulation of the apoptotic mitochondrial pathway. *Retrovirology* 5: 19
- Hulgan T, Samuels DC (2020) Mitochondria and HIV: A troubled relationship enters its fourth decade. *Clin Infect Dis*
- Hurwitz BE, Klimas NG, Llabre MM, Maher KJ, Skyler JS, Bilsker MS, McPherson-Baker S, Lawrence PJ, Laperriere AR, Greeson JM *et al* (2004) HIV, metabolic syndrome X, inflammation, oxidative stress, and coronary heart disease risk : role of protease inhibitor exposure. *Cardiovasc Toxicol* 4: 303-316
- Ivanov AV, Valuev-Elliston VT, Ivanova ON, Kochetkov SN, Starodubova ES, Bartosch B, Isaguliants MG (2016) Oxidative Stress during HIV Infection: Mechanisms and Consequences. *Oxid Med Cell Longev* 2016: 8910396
- Jones LE, Perelson AS (2007) Transient viremia, plasma viral load, and reservoir replenishment in HIV-infected patients on antiretroviral therapy. *J Acquir Immune Defic Syndr* 45: 483-493
- Kessing CF, Nixon CC, Li C, Tsai P, Takata H, Mousseau G, Ho PT, Honeycutt JB, Fallahi M, Trautmann L *et al* (2017) In Vivo Suppression of HIV Rebound by Didehydro-Cortistatin A, a "Block-and-Lock" Strategy for HIV-1 Treatment. *Cell Rep* 21: 600-611
- Masia M, Padilla S, Bernal E, Almenar MV, Molina J, Hernandez I, Graells ML, Gutierrez F (2007) Influence of antiretroviral therapy on oxidative stress

and cardiovascular risk: a prospective cross-sectional study in HIV-infected patients. *Clin Ther* 29: 1448-1455

Mastrantonio R, Cervelli M, Pietropaoli S, Mariottini P, Colasanti M, Persichini T (2016) HIV-Tat Induces the Nrf2/ARE Pathway through NMDA Receptor-Elicited Spermine Oxidase Activation in Human Neuroblastoma Cells. *PLoS One* 11: e0149802

Mondal D, Pradhan L, Ali M, Agrawal KC (2004) HAART drugs induce oxidative stress in human endothelial cells and increase endothelial recruitment of mononuclear cells: exacerbation by inflammatory cytokines and amelioration by antioxidants. *Cardiovasc Toxicol* 4: 287-302

Oexle H, Gnaiger E, Weiss G (1999) Iron-dependent changes in cellular energy metabolism: influence on citric acid cycle and oxidative phosphorylation. *Biochim Biophys Acta* 1413: 99-107

Opii WO, Sultana R, Abdul HM, Ansari MA, Nath A, Butterfield DA (2007) Oxidative stress and toxicity induced by the nucleoside reverse transcriptase inhibitor (NRTI)--2',3'-dideoxycytidine (ddC): relevance to HIV-dementia. *Exp Neurol* 204: 29-38

Pinti M, Biswas P, Troiano L, Nasi M, Ferraresi R, Mussini C, Vecchiet J, Esposito R, Paganelli R, Cossarizza A (2003) Different sensitivity to apoptosis in cells of monocytic or lymphocytic origin chronically infected with human immunodeficiency virus type-1. *Exp Biol Med (Maywood)* 228: 1346-1354

Pyo CW, Yang YL, Yoo NK, Choi SY (2008) Reactive oxygen species activate HIV long terminal repeat via post-translational control of NF-kappaB. *Biochem Biophys Res Commun* 376: 180-185

Ramratnam B, Ribeiro R, He T, Chung C, Simon V, Vanderhoeven J, Hurley A, Zhang L, Perelson AS, Ho DD *et al* (2004) Intensification of antiretroviral therapy accelerates the decay of the HIV-1 latent reservoir and decreases, but does not eliminate, ongoing virus replication. *J Acquir Immune Defic Syndr* 35: 33-37

Shytaj IL, Chirullo B, Wagner W, Ferrari MG, Sgarbanti R, Corte AD, LaBranche C, Lopalco L, Palamara AT, Montefiori D *et al* (2013) Investigational treatment suspension and enhanced cell-mediated immunity at rebound followed by drug-free remission of simian AIDS. *Retrovirology* 10: 71

Shytaj IL, Nickel G, Arts E, Farrell N, Biffoni M, Pal R, Chung HK, LaBranche C, Montefiori D, Vargas-Inchaustegui D *et al* (2015) Two-Year Follow-Up of Macaques Developing Intermittent Control of the Human Immunodeficiency Virus Homolog Simian Immunodeficiency Virus SIVmac251 in the Chronic Phase of Infection. *J Virol* 89: 7521-7535

Tasker C, Subbian S, Gao P, Couret J, Levine C, Ghanny S, Soteropoulos P, Zhao X, Landau N, Lu W *et al* (2016) IFN-epsilon protects primary macrophages against HIV infection. *JCI Insight* 1: e88255

Tyagi P, Pal VK, Agrawal R, Singh S, Srinivasan S, Singh A (2020) Mycobacterium tuberculosis Reactivates HIV-1 via Exosome-Mediated Resetting of Cellular Redox Potential and Bioenergetics. *mBio* 11

Valle-Casuso JC, Angin M, Volant S, Passaes C, Monceaux V, Mikhailova A, Bourdic K, Avettand-Fenoel V, Boufassa F, Sitbon M *et al* (2019) Cellular Metabolism Is a Major Determinant of HIV-1 Reservoir Seeding in CD4(+) T Cells and Offers an Opportunity to Tackle Infection. *Cell Metab* 29: 611-626 e615

Vernekar AA, Sinha D, Srivastava S, Paramasivam PU, D'Silva P, Mugesh G (2014) An antioxidant nanozyme that uncovers the cytoprotective potential of vanadia nanowires. *Nat Commun* 5: 5301

Weiss M, Kost B, Renner-Muller I, Wolf E, Mylonas I, Bruning A (2016) Efavirenz Causes Oxidative Stress, Endoplasmic Reticulum Stress, and Autophagy in Endothelial Cells. *Cardiovasc Toxicol* 16: 90-99

26th Jan 2021

Dear Prof. Singh,

Thank you for the submission of your revised manuscript to EMBO Molecular Medicine. We have now received the enclosed report from the two referees who were asked to re-assess it. As you will see the referees are overall supportive and I am pleased to inform you that we will be able to accept your manuscript pending the following amendments:

The referees still raised a series of concerns about your work, which need to be addressed and/or discussed.

1. Referee #2 suggested a couple of experiments to further improve the study - 1) assessment of the longer effects of Vs treatment in infected CD4+ cells; 2) analysis of gene expression change in primary cells. We would strongly encourage you to address Comment #1 to enhance the translational aspect of the study, but this is not mandatory for publication. Regarding Comment #2, if you have such data at hand, we would be happy for you to include it. Otherwise, please discuss this Referee's comment.

2. Please address all the other minor issues raised by both referees.

On a more editorial level:

1. In the main manuscript file, please do the following:

- remove the yellow color font
- callout for Figure 6E is missing, please fix it.
- Remove Appendix Table of Contents from main manuscript file.
- FUNDING: please add funding information to the manuscript file.
- In Materials and Methods (and in the checklist), include a statement that the experiments conformed to the principles set out in the WMA Declaration of Helsinki and the Department of Health and Human Services Belmont Report.

2. Appendix:

- Move Materials and Methods out of the Appendix into the main manuscript file.
- Please rename Appendix Tables 1A, 1B, and 1C to 1,2,3, and move them to the Appendix pdf. Please also update the callouts and nomenclature throughout.

3. Please add the ORCID ID to the account of Govindasamy Mugesh in our system. Please note that ORCID numbers can only be added by the individual account owners. Therefore, you would need to ask Dr. Mugesh to do it.

4. Checklist:

- please enter both co-corresponding authors' names, manuscript number, and journal name.

5. Please add a formal "Data Availability" section (placed after Materials & Method). Since this study does not generate large-scale datasets, please include the following sentence in this section- "This study includes no data deposited in external repositories".

6. For more information: There is space at the end of each article to list relevant web links for further

consultation by our readers. Could you identify some relevant ones and provide such information as well? Some examples are patient associations, relevant databases, OMIM/proteins/genes links, author's websites, etc...

7. We now encourage the publication of source data, particularly for electrophoretic gels, blots, but also microscopy images with the aim of making primary data more accessible and transparent to the reader. Would you be willing to provide a PDF file per figure that contains the original, uncropped, and unprocessed scans of all or key gels used in the figure (including molecular weight markers)? The PDF files should be labeled with the appropriate figure/panel number (1 file/figure), and should have molecular weight markers; further annotation may be useful but is not essential. The PDF files will be published online with the article as supplementary "Source Data" files. If you have any questions regarding this just contact me.

8. I have slightly modified and shortened the synopsis text. Please let me know if it is fine like this or if you would like to introduce further modifications.

This study described a vanadium pentoxide (V₂O₅)-based nanozyme that bolsters the anti-HIV potential of immune cells and suppresses viral rebound in latently infected CD4⁺ T cells derived from HIV subjects.

- Ultrathin sheets of V₂O₅-based nanozyme (Vs) exhibited efficient glutathione peroxidase activity inside the HIV infected cells.
- Vs suppressed HIV reactivation in cell line models of latency and in latently infected human CD4⁺ T cells.
- Vs inhibited HIV replication in primary CD4⁺ T cells and macrophages.
- Vs altered the expression of genes involved in the antioxidant response, HIV transcription, and apoptosis.
- Vs combined with antiretrovirals could be explored for delaying viral rebound, replication, and reseeding of reservoirs.

9. As part of the EMBO Publications transparent editorial process initiative (see our Editorial at <http://embomolmed.embopress.org/content/2/9/329>), EMBO Molecular Medicine will publish online a Review Process File (RPF) to accompany accepted manuscripts.

-In the event of acceptance, this file will be published in conjunction with your paper and will include the anonymous referee reports, your point-by-point response and all pertinent correspondence relating to the manuscript. Let us know if you do NOT agree with this.

I look forward to seeing a revised version of your manuscript as soon as possible.

Sincerely,
Jingyi

Please submit your revised manuscript within two weeks. I look forward to seeing a revised form of your manuscript as soon as possible.

I look forward to reading a new revised version of your manuscript as soon as possible.

Yours sincerely,

Jingyi Hou

Jingyi Hou
Editor
EMBO Molecular Medicine

*** Instructions to submit your revised manuscript ***

To submit your manuscript, please follow this link:

<https://embomolmed.msubmit.net/cgi-bin/main.plex>

- 1) a .docx formatted version of the manuscript text (including Figure legends and tables)
- 2) Separate figure files*
- 3) supplemental information as Expanded View and/or Appendix. Please carefully check the authors guidelines for formatting Expanded view and Appendix figures and tables at <https://www.embopress.org/page/journal/17574684/authorguide#expandedview>
- 4) a letter INCLUDING the reviewer's reports and your detailed responses to their comments (as Word file).
- 5) The paper explained: EMBO Molecular Medicine articles are accompanied by a summary of the articles to emphasize the major findings in the paper and their medical implications for the non-specialist reader. Please provide a draft summary of your article highlighting
 - the medical issue you are addressing,
 - the results obtained and
 - their clinical impact.This may be edited to ensure that readers understand the significance and context of the research.

Please refer to any of our published articles for an example.

6) For more information: There is space at the end of each article to list relevant web links for further consultation by our readers. Could you identify some relevant ones and provide such information as well? Some examples are patient associations, relevant databases, OMIM/proteins/genes links, author's websites, etc...

7) Author contributions: the contribution of every author must be detailed in a separate section.

8) EMBO Molecular Medicine now requires a complete author checklist (<https://www.embopress.org/page/journal/17574684/authorguide>) to be submitted with all revised manuscripts. Please use the checklist as guideline for the sort of information we need WITHIN the manuscript. The checklist should only be filled with page numbers where the information can be found. This is particularly important for animal reporting, antibody dilutions (missing) and exact values and n that should be indicated instead of a range.

9) Every published paper now includes a 'Synopsis' to further enhance discoverability. Synopses are displayed on the journal webpage and are freely accessible to all readers. They include a short stand first (maximum of 300 characters, including space) as well as 2-5 one sentence bullet points that summarise the paper. Please write the bullet points to summarise the key NEW findings. They should be designed to be complementary to the abstract - i.e. not repeat the same text. We encourage inclusion of key acronyms and quantitative information (maximum of 30 words / bullet point). Please use the passive voice. Please attach these in a separate file or send them by email, we will incorporate them accordingly.

You are also welcome to suggest a striking image or visual abstract to illustrate your article. If you do please provide a jpeg file 550 px-wide x 400-px high.

10) A Conflict of Interest statement should be provided in the main text

11) Please note that we now mandate that all corresponding authors list an ORCID digital identifier. This takes <90 seconds to complete. We encourage all authors to supply an ORCID identifier, which will be linked to their name for unambiguous name identification.

Currently, our records indicate that the ORCID for your account is 0000-0001-6761-1664.

Link Not Available

12) The system will prompt you to fill in your funding and payment information. This will allow Wiley to send you a quote for the article processing charge (APC) in case of acceptance. This quote takes into account any reduction or fee waivers that you may be eligible for. Authors do not need to pay any fees before their manuscript is accepted and transferred to our publisher.

Photos 400-800 DPI

*Additional important information regarding figures and illustrations can be found at <https://bit.ly/EMBOPressFigurePreparationGuideline>

The system will prompt you to fill in your funding and payment information. This will allow Wiley to send you a quote for the article processing charge (APC) in case of acceptance. This quote takes into account any reduction or fee waivers that you may be eligible for. Authors do not need to pay any fees before their manuscript is accepted and transferred to our publisher.

***** Reviewer's comments *****

Referee #2 (Comments on Novelty/Model System for Author):

The revised manuscript contains relevant cellular models, but the biggest improvement is represented by the addition of experiments performed on samples obtained from HIV-1 infected individuals on antiretroviral therapy.

Referee #2 (Remarks for Author):

The Revised manuscript has significantly improved by the addition of experiments on primary CD4 T cell model of infection and in particular with the experiment on CD4 T cells from HIV-1 infected individuals on ART.

My comments to the revised version are:

In their experiment, infected CD4+ cells were followed at 3 and 5 days with or without Vs treatment. I am wondering what happens with the viral replication at longer times in the presence of Vs? What will happen if the intracellular GSH is consumed? Will the viral latency be established in any case, just with the delayed kinetics think this information would be very useful to understand the potentials of Vs as a therapeutic agent. What will happen at 7 or 14 or even 21 days post infection?

Also, it would be useful to know what happens with gene expression patterns in primary cells. Ideally, this information could have been retrieved from patient samples, but that will be difficult to respond. On the other hand, the similar experiment on in vitro cells should have been performed to complete the picture ie our understanding on the possible mechanisms.

Minor - row 316 the authors mention 123 genes which show differential expression (DE) - how did they come up with these genes, as according to my counts there are 118 of DE genes.

Referee #3 (Remarks for Author):

Authors have addressed my concerns. The manuscript has improved significantly. I only have few

minor comments.

1. Line 26, replace "proliferation" with "replication."
2. Line 34, replace "multiplication" with "replication."
3. It should read "RT-qPCR" not "qRT-PCR."
4. Cell tropism for HIV is an old fashion term. In Line 369, authors should replace "T cell-tropic virus" with CXCR4-using virus. Similarly, in Line 379, "macrophage tropic virus" should be replaced with CCR5-using virus. Please make corrections throughout the manuscript.

Manuscript number: EMM-2020-13314

Title: Antioxidant nanozyme counteracts HIV-1 by modulating intracellular redox potential

Dear Dr. Hou,

We thank the reviewers for responding positively to our manuscript **“Antioxidant nanozyme counteracts HIV-1 by modulating intracellular redox potential”**. Based on the reviewers' and editor's comments, we have revised the manuscript. A point-by-point response to the reviewers' comments is given below:

Editorial changes

We have made all the changes in the main manuscript, appendix, checklist, Data availability, and source data as per the editor's email.

Reviewer #1

The Revised manuscript has significantly improved by the addition of experiments on primary CD4 T cell model of infection and in particular with the experiment on CD4 T cells from HIV-1 infected individuals on ART.

My comments to the revised version are:

In their experiment, infected CD4+ cells were followed at 3 and 5 days with or without Vs treatment. I am wondering what happens with the viral replication at longer times in the presence of Vs? What will happen if the intracellular GSH is consumed? Will the viral latency be established in any case, just with the delayed kinetics think this information would be very useful to understand the potentials of Vs as a therapeutic agent. What will happen at 7 or 14 or even 21 days post infection?

We thank the reviewer for this comment. However, we have addressed this concern in a more clinically relevant primary cell model system. We assessed the long-term influence of Vs on viral latency and rebound in CD4⁺ T cells isolated from HIV-infected subjects on suppressive antiretroviral therapy (ART). We exposed CD4⁺ T cells with antiretrovirals (ARVs) alone or in combination with Vs (Vs + ARVs) for 21 days. We showed that the viral RNA levels from all three subjects' cells exposed to Vs + ARVs were below detection limit compared to only one subject's cells in the case of ARVs alone (Figure 7B in the revised manuscript). These results indicate a better ability of Vs + ARVs to inhibit virus transcription and raise the possibility of including Vs with the frontline treatment for faster suppression.

The antioxidant glutathione (GSH) is abundantly present in mammalian cells (1- 10 mM) (1), and depletion of GSH adversely affects cell viability and induces apoptosis (2). We have shown that exposure to Vs in the primary

cells is not inducing apoptosis (Figure EV5B in the revised manuscript). Also, we do not think that long-term exposure to Vs is depleting GSH in the primary cells. We have shown that the total HIV DNA content between the freshly isolated patient's CD4⁺ T cells and the expanded cells treated with ARVs alone or Vs + ARVs remains comparable for the entire duration (21 days) of the experiment (Figure 7E in the revised manuscript). Data suggest that prolonged exposure to Vs is not selectively inducing loss of HIV infected cells. To further address this point and enhance the translational aspect of Vs, we have data at hand showing the viability status of the patients' CD4⁺ T cells treated with ARVs alone or Vs + ARVs for 14 and 28 days using live-dead stain. As expected, we found no difference in the fraction of viable cells in the case ARVs alone or Vs + ARVs at 14 and 28 days post-treatment (Fig. 1 [Appendix Figure S8 in the revised manuscript]). We have included this data in the revised manuscript. These findings indicate the excellent potential of Vs as a therapeutic agent in combination with current ART regimens.

Fig 1. Viability of primary CD4⁺ T cells is not affected by Vs treatment.

Primary CD4⁺ T cells isolated from ARV-suppressed individuals were culture in presence of ARV alone or Vs + ARV (*as mentioned in Materials and Methods*). At day 14 and day 28 ARV and ARV+Vs treatment, cells were stained with Aqua Dead Cell staining dye and cell viability was analysed using flow cytometry. Data are representative of three donors. ns – non-significant by 2-way ANOVA with Sidak's multiple comparison.

Also, it would be useful to know what happens with gene expression patterns in primary cells. Ideally, this information could have been retrieved from patient samples, but that will be difficult to respond. On the other hand, the similar experiment on in vitro cells should have been performed to complete the picture ie our understanding on the possible mechanisms.

We do not have such data in primary cells. However, we have RT-qPCR data of four Nrf2-dependent antioxidant genes (GPX1, GPX4, GSR and SRXN1) in

latently infected human CD4⁺ T cells incubated with ARVs alone or Vs + ARVs for 21 days. Consistent with our findings in the U1 cell line, Vs treatment represses the expression of antioxidant genes in the patient CD4⁺ T cells (Fig. 2 [Appendix Figure S7 in the revised manuscript]). We have included this data and appropriately discussed it in the revised manuscript.

Fig 2. Expression of antioxidant genes in primary CD4⁺ T cells are lowered upon Vs exposure. ARV or ARV+Vs treated cells were cultured for 21 days under standard conditions (*see Materials and Methods*). 21 days post culturing, RNA was isolated and expression of antioxidant genes, GPX1, GPX4, GSR, and SRXN1 were analysed by RT-qPCR. Data are representative of three donors. ** P<0.01, * P<0.05, ns – non-significant by 2-way ANOVA with Sidak's multiple comparison.

Minor - row 316 the authors mention 123 genes which show differential expression (DE) - how did they come up with these genes, as according to my counts there are 118 of DE genes.

We have corrected the number of genes to 118. We apologize for this confusion.

Reviewer # 2

1. Line 26, replace "proliferation" with "replication."
2. Line 34, replace "multiplication" with "replication."
3. It should read "RT-qPCR" not "qRT-PCR."
4. Cell tropism for HIV is an old fashion term. In Line 369, authors should replace "T cell-tropic virus" with CXCR4-using virus. Similarly, in Line 379, "macrophage tropic virus" should be replaced with CCR5-using virus. Please make corrections throughout the manuscript.

We have taken care of the above aspects as per the reviewer's suggestion.

References

1. Gutscher, M., Pauleau, A. L., Marty, L., Brach, T., Wabnitz, G. H., Samstag, Y., Meyer, A. J., and Dick, T. P. (2008) Real-time imaging of the intracellular glutathione redox potential. *Nat Methods* **5**, 553-559
2. De Nicola, M., and Ghibelli, L. (2014) Glutathione depletion in survival and apoptotic pathways. *Front Pharmacol* **5**, 267

19th Feb 2021

Dear Amit,

We are pleased to inform you that your manuscript is accepted for publication and is now being sent to our publisher to be included in the next available issue of EMBO Molecular Medicine.

We would like to remind you that as part of the EMBO Publications transparent editorial process initiative, EMBO Molecular Medicine will publish a Review Process File online to accompany accepted manuscripts. If you do NOT want the file to be published or would like to exclude figures, please immediately inform the editorial office via e-mail.

Please read below for additional IMPORTANT information regarding your article, its publication and the production process.

Congratulations on your interesting work,

Jingyi

Jingyi Hou
Editor
EMBO Molecular Medicine

Follow us on Twitter @EmboMolMed
Sign up for eTOCs at embopress.org/alertsfeeds

***** Reviewer's comments *****

*** ** IMPORTANT INFORMATION *** **

SPEED OF PUBLICATION

The journal aims for rapid publication of papers, using the advance online publication "Early View" to expedite the process: A properly copy-edited and formatted version will be published as "Early View" after the proofs have been corrected. Please help the Editors and publisher avoid delays by providing e-mail address(es), telephone and fax numbers at which author(s) can be contacted.

Should you be planning a Press Release on your article, please get in contact with embomolmed@wiley.com as early as possible, in order to coordinate publication and release dates.

LICENSE AND PAYMENT:

All articles published in EMBO Molecular Medicine are fully open access: immediately and freely

available to read, download and share.

EMBO Molecular Medicine charges an article processing charge (APC) to cover the publication costs. You, as the corresponding author for this manuscript, should have already received a quote with the article processing fee separately. Please let us know in case this quote has not been received.

Once your article is at Wiley for editorial production you will receive an email from Wiley's Author Services system, which will ask you to log in and will present you with the publication license form for completion. Within the same system the publication fee can be paid by credit card, an invoice, pro forma invoice or purchase order can be requested.

Payment of the publication charge and the signed Open Access Agreement form must be received before the article can be published online.

PROOFS

You will receive the proofs by e-mail approximately 2 weeks after all relevant files have been sent to our Production Office. Please return them within 48 hours and if there should be any problems, please contact the production office at embopressproduction@wiley.com.

Please inform us if there is likely to be any difficulty in reaching you at the above address at that time. Failure to meet our deadlines may result in a delay of publication.

All further communications concerning your paper proofs should quote reference number EMM-2020-13314-V3 and be directed to the production office at embopressproduction@wiley.com.

Thank you,

Jingyi Hou
Editor
EMBO Molecular Medicine

Corresponding Author Name: Govindasamy Mugesh and Amit Singh

Manuscript Number: EMM-2020-13314